# Resilient Coresets and Clustering

**MohammadHossein Bateni** [1]   **Silvio Lattanzi** [1]   **Morteza Monemizadeh** [2]   **Ashkan Norouzi-Fard** [1]

## Abstract

Many machine learning problems are geometric at their core, relying on metric representations of data for tasks such as clustering, prototype selection, nearest-neighbor search, and graph-based learning. Furthermore, data is constantly evolving and it is routinely transformed through dimensionality reduction, random projections, feature embeddings, compression, or privacy-preserving mechanisms. These transformations are designed to preserve geometry approximately. As a result, they preserve objective values for many geometric optimization problems, but they fail to guarantee that algorithmic outcomes remain consistent. In this work, we study *resilient data summaries* for geometric optimization. Building on the notion of $\gamma$-*resilient algorithms* from (Ahmadian et al., 2024), we introduce $\gamma$-*resilient coresets*. A $\gamma$-resilient $(k, \varepsilon)$-coreset is a compact, weighted summary that guarantees a $(1 + \varepsilon)$ approximation to the objective and enforces stability at the level of assignments. We complement our positive result with a lower bound showing that to obtain a tight approximation for resilient clustering it is necessary to use a bi-criteria solution.

## 1. Introduction

Many core problems in machine learning are fundamentally geometric. Datasets are represented as points in a metric space, where distances encode similarity, dissimilarity, or semantic relationships between entities. This geometric perspective underlies a wide range of tasks, including clustering and representation learning, facility selection, nearest-neighbor search, and cut problems derived from similarity-based embeddings. In these contexts, the geometry of the data shapes both the quality of the objective and the structure of the learned solution, such as cluster assignments, selected representatives, or partitions of data.

Modern machine learning pipelines rarely operate on a single, fixed representation of the data. Instead, raw data naturally evolves over time (e.g., location data) or is routinely transformed through dimensionality reduction, random projections, feature embeddings, compression, quantization, or privacy-preserving mechanisms. These transformations aim to approximately preserve the underlying geometric structure while improving computational efficiency, memory usage, or robustness. Consequently, the same dataset may appear in multiple representations that are not identical, yet are intended to encode essentially the same metric information. Examples include projecting high-dimensional embeddings into a lower-dimensional space for efficient clustering, sketching or hashing features for scalable nearest-neighbor search, and federated or distributed learning where local updates induce small perturbations to a global dataset.

To formalize this phenomenon, Ahmadian et al. (2024) introduced the notions of $\xi$-*close point sets* and $\gamma$-*resilient algorithms*. Two point sets $P$ and $P'$ are $\xi$-close if there exists a bijection between them that preserves all pairwise distances within a multiplicative factor of $1 \pm \xi$. This notion provides a precise formalization of near-isometric perturbations: the two datasets represent essentially the same geometry, up to small, potentially consequential distortions.

A randomized algorithm is $\gamma$-resilient (where $\gamma$ is a function of $\xi$) if, when executed independently on two $\xi$-close datasets using shared randomness, its outputs differ on at most a $\gamma(\xi)$ fraction of output in expectation.[1] Unlike classical approximation guarantees, which control only objective values, $\gamma$-resilience enforces pointwise agreement, directly measuring solution stability. Intuitively, this concept captures whether discrete algorithmic decisions vary smoothly with the underlying metric, and whether different representations of the same data lead to nearly identical outputs.

---

[1]Google Research [2]Eindhoven University of Technology. Correspondence to: Ashkan Norouzi-Fard <ashkan-norouzi@google.com>.

*Proceedings of the 43rd International Conference on Machine Learning*, Seoul, South Korea. PMLR 306, 2026. Copyright 2026 by the author(s).

---

[1]A natural motivation for the resiliency property comes from the imprecision inherent in real-world measurements. Consider GPS data as an example: given a ground-truth point set $P_1$, exact measurement is challenging in practice, and we instead observe a noisy point set $P_2$ (due to measurement errors). In such settings, a resilient data summary on $P_2$ ensures that we recover a data summary similar to the one we would have obtained from the (inaccessible) $P_1$.

**Our Contributions.** In this work, we extend this framework from individual algorithms to *coresets*. A coreset is a small, weighted summary of a dataset that approximately preserves the objective for a class of optimization problems. Coresets are widely used in scalable machine learning, streaming algorithms, and distributed computation. They enable fast approximations with provable guarantees.

We introduce $\gamma$-*resilient* $(k, \varepsilon)$-*coresets*, which simultaneously guarantee $(1 + \varepsilon)$-approximation of the objective and stable assignments. That is, when constructed on two $\xi$-close datasets using shared randomness, the coreset representatives and the assignment of points to coreset representatives differ on at most a $\gamma(\xi)$ fraction of the output in expectation. In this way, resilience becomes a property of the summary itself, rather than of any particular algorithm.

Achieving $\gamma$-resilience requires careful handling of randomness and discrete decisions. Specifically:

**(1) Randomness:** Shared randomness aligns the random choices across executions on different $\xi$-close inputs, ensuring that observed differences arise solely from geometric perturbations rather than independent randomness. However, shared randomness alone is not sufficient.

**(2) Discrete decisions and coupling:** Even with perfectly aligned random bits, small metric distortions can still cause discrete decisions (comparisons, threshold crossings, point-to-center assignments) to diverge. To reason rigorously about this, we employ the notion of a *coupling*, which defines a joint probabilistic execution of the algorithm on both inputs. Coupling coordinates the processing of corresponding points so that their computations are as aligned as possible. This provides a framework to "track" points under perturbations. In essence, coupling here is a formal tool to control the divergence of discrete decisions under input perturbations, ensuring that discrete outputs of the algorithm are stable except for a small fraction of points.

In this way, $\gamma$-resilience can be interpreted as the existence of such a coupling under which all but a $\gamma(\xi)$ fraction of points follow identical computational paths and are assigned consistently across $\xi$-close datasets.

In addition to our positive result we also address an open question in (Ahmadian et al., 2024) by providing a new lower bound for the resilient clustering problem. Informally, our lower bound shows that for the $k$-center, $k$-median and $k$-means problems, asking for resiliency makes the clustering problem significantly harder. We do this by presenting a lower bound for the problem even when we consider the problem on a single dimension and when we allow the algorithm to be bi-criteria (and open many more centers)[2].

---

[2]Our results are also applies to coreset, since any coreset algorithm is also a bi-criteria algorithm. They are equivalent from lower bound point of view.

This essentially justifies previous work on bi-criteria algorithm for the problems and shows that our coreset solution achieves a very good trade-off between approximation quality, resiliency and the number of centers used.

**Overview and intuition.** We present a randomized construction of *resilient coresets* for geometric clustering, focusing on the $k$-median and $k$-means problems in constant dimension. Our goal is to compute a small weighted subset $S \subseteq P$ such that (i) $S$ approximates the clustering cost of the original point set $P$ within a $1 \pm \epsilon$ factor for every choice of $k$ centers, and (ii) the coreset remains stable under small perturbations of the input. Formally, if $P'$ is $\xi$-close to $P$ under a near-isometric bijection, then the coresets constructed from $P$ and $P'$ differ in only a small fraction of point-to-representative assignments. This resiliency requirement rules out many standard coreset constructions, since even small perturbations can cause large changes in sampling outcomes or partition boundaries.

Our $\gamma$-resilient coreset construction is inspired by the grid-based framework of (Frahling & Sohler, 2005), but differs substantially in both design and analysis due to the additional consistency constraints imposed by resiliency. At a high level, the algorithm combines hierarchical grid decompositions with carefully injected randomness to simultaneously control approximation and resiliency errors. We first apply a random shift to the point set, ensuring that point locations relative to grid boundaries are uniformly random. We then construct a quadtree-style hierarchy of grids at exponentially decreasing resolutions and classify cells as *heavy* or *light* based on their contribution to the clustering cost. Heavy cells are recursively refined, while the remaining light cells form a partition $\Lambda$ of the space.

A natural approach would be to select a representative for each light cell $c \in \Lambda$ and assign all points in $c$ to this representative. While this yields a valid $(k, \epsilon)$-coreset, it fails to satisfy the $\gamma$-resiliency requirement: under $\xi$-close perturbations, both the representative and the other points in $c$ may move across cell boundaries, making the induced assignments unstable. To overcome this challenge, our key idea is to assign points from the light cells of $\Lambda$ to sampled representatives drawn from their ancestor cells in a carefully chosen set of random grids, which we refer to as *pivot grids*. Each representative in a pivot grid aggregates and represents points from multiple light cells in its subtree up to the depth $O(\log(1/\varepsilon))$ recursively, and its weight is set to the total number of points assigned to it.

The analysis relies on two complementary components. Regarding the approximation, Lemmas 3.2 and 3.3 show that assigning points from light cells to representatives in pivot grids introduces only a small relative error relative to the clustering cost. For resiliency, we identify two potential

sources of error. The first is that two points $p, q \in P$ lying in the same pivot grid cell may have images $\pi(p), \pi(q) \in P'$ that fall into different cells. The second is when the image of a coreset representative selected in $P$ may move to a different pivot grid cell in $P'$, leading to changes in point-to-representative assignments. To address the first issue, we establish Lemmas 3.4 and 3.5, which together bound the probability that such pairs of points are separated under perturbations. These bounds are combined in Lemma 3.6 to control this type of resiliency error. To handle the second issue, Lemma 3.8 bounds the expected number of representatives whose images move to different pivot grid cells.

Our hardness results further illustrate the strength of our proposed coreset. Specifically, we construct an instance in 1-dimensional space demonstrating that any constant-approximation algorithm—even one permitted to open a constant factor more centers, e.g., $100k$—cannot achieve a resiliency factor better than $\Omega(\ln(1 + \xi)/\xi)$, where $0 < \xi \leq 1$. In contrast, a special case of our coreset achieves a $(1 + \epsilon)$-approximate solution by opening $O(\log n(\log n + k) \cdot \text{poly}(\frac{1}{\varepsilon}))$ centers, while maintaining $O(\ln(1 + \xi))$ resiliency, where $n$ denotes the number of points in the input instance. Intuitively, our algorithm achieves significantly better approximation guarantee and resilience, while opening extra centers. This hardness results is presented for $k$-center, $k$-median, and $k$-means problems.

The hardness result comprises several key components. We first define a family of hard instances wherein the base instance consists of $k$ well-separated clusters; any constant-approximation solution is thus forced to open at least one center in almost all of these clusters. Subsequently, we generate a set of instances by perturbing these clusters; bringing some closer together and partitioning others. Notably, these movements are highly constrained, as the distance between any two points can only change by a factor of $1 + \xi$. By carefully shifting the points, we construct a chain of closely related instances. We then demonstrate that any deterministic algorithm must significantly alter its clustering between the base instance and the instance at the end of each chain. Finally, we extend this result to randomized algorithms via Yao's Minimax Principle. The formal structure and proof is presented in Section 4.

**Related work.** The most closely related work is (Ahmadian et al., 2024) were the notions of $\xi$-*close point sets* and $\gamma$-*resilient algorithms* are introduced. This work uses the definition and extends it to the notion of coresets. Our coresets result can also be seen in their setting as a better bi-criteria approximation algorithm for the Euclidean space. In addition, we also provide a new lower bound for the general clustering problem answering an open question in (Ahmadian et al., 2024).

In addition, our work is also closely related to work on (average) sensitivity, consistent clustering, perturbation resilience, stability and robustness that we discuss next.

*Sensitivity and average sensitivity.* Sensitivity and average sensitivity have been extensively studied for dynamic programming, clustering, graph and learning problems (Yoshida & Zhou, 2020; Varma & Yoshida, 2021; Kumabe & Yoshida, 2022a; Yoshida & Ito, 2022; Hara & Yoshida). This notion focuses on the impact of removing/adding a *single* point from/to the input, in contrast to the resilient setting which lets *all* the points to change their positions by a limited amount. Kumabe & Yoshida (2022b) introduces the notion of Lipschitz continuity for graph problems and designs algorithms in this setting.

*Consistent clustering.* A related area of research is *consistent* clustering (Lattanzi & Vassilvitskii, 2017; Chan et al., 2018; Cohen-Addad et al., 2019; Jaghargh et al., 2019; Fichtenberger et al., 2021; Guo et al., 2021). In this setting, the input evolves over time and the goal is to maintain a good clustering with as few changes as possible. Unfortunately the results in this interesting setting do not extend in the presence of input perturbations.

*Perturbation resilience, stability and robustness.* An instance has $(\alpha, \epsilon)$-perturbation resilience if the optimal clustering continues to remain optimal even when all pairwise distances by a factor of at most $\alpha$ for all but an $\epsilon$ fraction of the points. In a series of papers (Bilu & Linial, 2012; Balcan et al., 2020; Agarwal et al., 2020; Chekuri & Gupta, 2018), several algorithms have been presented in this setting. Similarly there are several influential papers studying different notions of stability or robustness for clustering and providing better algorithms under certain stability assumption on the optimal solution (Awasthi et al., 2012; Lattanzi et al., 2015; Balcan et al., 2014; Chakrabarty & Negahbani, 2023). Although close, it is important to note that the focus of these papers is different from ours as they are not interested in providing any guarantee on resilience to adversarial perturbation in the metric space.

**Outline of the paper.** In Section 2, we introduce the notations and definitions used in this paper. In Section Section 3, we present our main algorithmic result a $\gamma$-*resilient* $(k, \varepsilon)$-*coresets*. In Section Section 4, we establish our hardness result. In Sections Appendix C and Appendix D, we present a general lower bound for resilient clustering problems. In Appendix A and Appendix B, we present missing proofs.

## 2. Preliminaries

In Euclidean $k$-median clustering, we are given a point set $P \subset \mathbb{R}^d$ and an integer $k$. The objective is to

compute a set $C^* \subset \mathbb{R}^d$ of $k$ centers that minimizes $\mathrm{cost}(P, C^*) = \sum_{p \in P} \mathrm{dist}(p, C^*)$, where $\mathrm{dist}(p, C^*) = \min_{c \in C^*} \mathrm{dist}(p, c)$ denotes the Euclidean distance from $p$ to its nearest center in $C^*$. We denote the optimal $k$-median cost of $P$ by $\mathrm{OPT}(P, k)$, and when $(P, k)$ are clear from context, we simply write $\mathrm{OPT}$.

The optimal clustering induced by the center set $C^* = \{c_1^*, \ldots, c_k^*\}$ is the partition $\{\mathcal{C}(c_1^*), \ldots, \mathcal{C}(c_k^*)\}$, where each cluster $\mathcal{C}(c_i^*)$ consists of all points that are strictly closer to $c_i^*$ than to any other center in $C^*$. If a point is equidistant to multiple centers, it may be assigned to any one of them.

Now suppose we have an (approximate or exact) $k$-median algorithm $\mathcal{A}$ whose output is a center set $C = \{c_1, \ldots, c_k\}$ and a corresponding clustering $\{\mathcal{C}_1, \ldots, \mathcal{C}_k\}$. For any point $p \in P$, we denote by $C(p)$ the center to which $p$ is assigned by $\mathcal{A}$. We redefine the output of the algorithm as

$$\mathcal{A}(P, k) = \{(p_1, C(p_1)), (p_2, C(p_2)), \ldots, (p_n, C(p_n))\},$$

where each pair $(p_j, C(p_j))$ records a point $p_j \in P$ together with its assigned center.

Ahmadian et al. (2024) defines the notions of $\epsilon$-*close* point sets and *resilient clustering*. We recall these definitions in a form adapted to our setting. Each point $p \in P$ is assumed to have a unique identifier, which we denote simply by $p$.

**Definition 2.1** ($\xi$-close point sets (Ahmadian et al., 2024)). Let $0 < \xi \le 1$, and let $P, P' \subset \mathbb{R}^d$ be point sets with a bijective mapping $\pi : P \to P'$. The sets $P$ and $P'$ are $\xi$-*close* if for all $p, q \in P$ with $\mathrm{dist}(p, q) > 0$, we have

$$(1-\xi)\,\mathrm{dist}(p, q) \le \mathrm{dist}(\pi(p), \pi(q)) \le (1+\xi)\,\mathrm{dist}(p, q),$$

and if $\mathrm{dist}(p, q) = 0$ then $\mathrm{dist}(\pi(p), \pi(q)) = 0$.

For any two sets $A$ and $B$, we define their symmetric difference as $A \triangle B = (A \setminus B) \cup (B \setminus A)$. We now define $\gamma$-resilient clustering.

**Definition 2.2** ($\gamma$-resilient algorithm (Ahmadian et al., 2024)). An algorithm $\mathcal{A}$ is a $\gamma$-*resilient $k$-median algorithm* (where $\gamma$ is a function of $\xi$) if for any $\xi$-close point sets $P$ and $P'$ with bijection $\pi : P \to P'$, the outputs satisfy

$$\mathbb{E}[\,|\pi(\mathcal{A}(P, k)) \triangle \mathcal{A}(P', k)|\,] \le \gamma(\xi)\,|P|,$$

where the expectation is over the random bits of $\mathcal{A}$ and where we denote the point $(\pi(p), \pi(C(p)))$ with $\pi(p, C(p))$, and hence the short-hand $\pi(\mathcal{A}(P, k)) = \{(\pi(p_1), \pi(C(p_1))), \ldots, (\pi(p_n), \pi(C(p_n)))\}$.

A key concept introduced in this paper is $\gamma$-*resilient $(k, \varepsilon)$-coresets*. We begin by recalling the definition of a $(k, \varepsilon)$-coreset (Har-Peled & Mazumdar, 2004; Frahling & Sohler, 2005) for $k$-median clustering. Given a point set $P \subset \mathbb{R}^d$, a weighted set $S \subset \mathbb{R}^d$ is a $(k, \varepsilon)$-*coreset* for $P$ if, for any set of $k$ centers $C \subset \mathbb{R}^d$, the $k$-median cost of $S$ approximates that of $P$ within a factor of $1 \pm \varepsilon$, i.e.,

$$(1 - \varepsilon)\,\mathrm{cost}(P, C) \le \mathrm{cost}(S, C) \le (1 + \varepsilon)\,\mathrm{cost}(P, C),$$

where $\mathrm{cost}(S, C) = \sum_{s \in S} w(s)\mathrm{dist}(s, C)$ and $w(s)$ is the weight associated with each $s \in S$.

Many coreset constructions (Har-Peled & Mazumdar, 2004; Frahling & Sohler, 2005; Chen, 2009) further provide a bijection $\phi : P \to S$ that maps each input point to a corresponding point in the coreset. We assume access to a coreset construction algorithm $\mathcal{A}$ that, given $P$, outputs a $(k, \varepsilon)$-coreset $S$ together with the bijection $\phi_{\mathcal{A}} : P \to S$. In this setting, we can represent the output of $\mathcal{A}$ as

$$\mathcal{A}(P, k) = \{(p_1, \phi_{\mathcal{A}}(p_1)), \ldots, (p_n, \phi_{\mathcal{A}}(p_n))\}.$$

**Definition 2.3** ($\gamma$-resilient $(k, \varepsilon)$-coreset). Let $0 < \xi, \gamma \le 1$ be two paramters. An algorithm $\mathcal{A}$ produces a $\gamma$-*resilient* $(k, \varepsilon)$-*coreset* if, for any $\xi$-close point sets $P$ and $P'$ with bijection $\pi : P \to P'$, the outputs satisfy

$$\mathbb{E}\Big[\,|\pi(\mathcal{A}(P, k)) \triangle \mathcal{A}(P', k)|\,\Big] \le \gamma(\xi)\,|P|,$$

where the expectation is over random choices of the algorithm $\mathcal{A}$. We use the short-hand $\pi(\mathcal{A}(P, k)) = \{(\pi(p_1), \phi_{\mathcal{A}}(\pi(p_1))), \ldots, (\pi(p_n), \phi_{\mathcal{A}}(\pi(p_n)))\}$, i.e., $\pi$ is applied to points and their assigned coreset representatives.

Observe that in $\pi(\mathcal{A}(P, k))$, each pair $(\pi(p_j), \phi_{\mathcal{A}}(\pi(p_j)))$ represents the coreset representative assigned to the image of $p_j$ under the mapping $\pi$, applying $\pi$ first and then $\phi_{\mathcal{A}}$.

## 3. Resilient Coreset Construction

We consider a point set $P \subseteq [\Delta]^d$ of size $|P| = n$, where $[\Delta]^d = \{1, \ldots, \Delta\}^d$ denotes the discrete grid and $\Delta = n^{O(1)}$. The problem is parameterized by $k$, the number of clusters in the $k$-median objective, and by an accuracy parameter $\varepsilon$. Our $\gamma$-resilient coreset construction is inspired by the framework of (Frahling & Sohler, 2005) and applies to $k$-median and $k$-means clustering in constant dimension. Nonetheless, our construction and its analysis differ significantly from theirs, due to the consistency constraints that we must preserve in our setting.

We first assume that the optimal $k$-median cost OPT is known. Our resilient coreset construction, denoted RESILIENTCORESET, consists of four steps.

**Step 1: Random shift.** We shift the point set $P$ by a uniformly random vector $v \in [0, \Delta]^d$. For each point $p \in P$, let $p + v$ denote its translation by $v$. The translated point set lies in $[0, 2\Delta]^d$, and from now on we identify each point $p$ with its translation $p + v$. The random vector $v$ is the one that is shared across all $\xi$-close point sets.

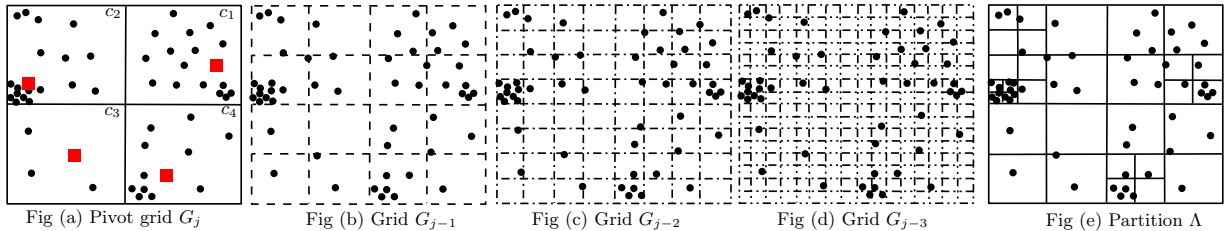

| Fig (a) Pivot grid $G_j$ | Fig (b) Grid $G_{j-1}$ | Fig (c) Grid $G_{j-2}$ | Fig (d) Grid $G_{j-3}$ | Fig (e) Partition $\Lambda$ |

*Figure 1.* In this figure, Fig. (a) shows the pivot grid $G_j$, while Figs. (b), (c), and (d) illustrate the non-pivot grids $G_{j-1}$, $G_{j-2}$, and $G_{j-3}$, respectively. Fig. (e) depicts the partition $\Lambda$, which consists of light cells from the grids $G_{j-1}$, $G_{j-2}$, and $G_{j-3}$. In Fig. (a), the red squares denote coreset representatives sampled uniformly at random from each non-empty cell of the pivot grid $G_j$. The pivot grid contains four cells, denoted by $c_1$, $c_2$, $c_3$, and $c_4$. All points lying in the light cells of $\Lambda$—which belong to the non-pivot grids $G_{j-1}$, $G_{j-2}$, and $G_{j-3}$—are assigned to the coreset representative corresponding to their ancestor cell in the pivot grid, that is, the unique cell in $\{c_1, c_2, c_3, c_4\}$ that contains them. For illustration, if $y = 4$, the next pivot grid would be $G_{j-4}$, which is not shown in the figure.

**Step 2: Pivot and non-pivot grids.** We impose a hierarchical family of grids $G_0, G_1, \ldots, G_{\log \Delta}, G_{\log(2\Delta)}$ over the domain $[2\Delta]^d$, where each cell $c \in G_i$ has side length $\ell_c = 2^i$. For a cell $c$, let $n_c = |c \cap P|$ denote the number of points of $P$ contained in $c$. See Figure 1 for an illustration.

We next define a collection $\mathcal{G}$ of *pivot grids*. Consider the grids $G_{\log(2\Delta)}, G_{\log(2\Delta)-1}, \ldots, G_{\log(2\Delta)-y}$, where $y = \log(a/\varepsilon)$ and the constant $a > 0$ will be fixed in the analysis. We select one of these grids uniformly at random; let the chosen grid be $G_j$, where $j \in \{\log(2\Delta), \ldots, \log(2\Delta) - y\}$.

From $G_j$, we form a sequence of finer grids by taking strides of size $y$. Specifically, we define $\mathcal{G} = \{G_j, G_{j-y}, G_{j-2y}, \ldots, G_{j-xy}\}$, where $x$ is the largest integer such that $j - xy \geq y$. This set is a set of grids forming an arithmetic progression with initial term $j$ and common difference $-y$. We refer to the grids in $\mathcal{G}$ as the *pivot grids*. The rest of the grids in the set $\{G_{\log(2\Delta)}, G_{\log \Delta}, \ldots, G_1, G_0\} \backslash \mathcal{G}$ are *non-pivot grids*. As an example, in Figure 1 for $y = 4$, the pivot grids are $G_j, G_{j-4}, G_{j-8}, G_{12}, \ldots$, while the non-pivot grids are $G_{j-1}, G_{j-2}, G_{j-3}, G_{j-5}, G_{j-6}, G_{j-7}, G_{j-9}, \ldots$.

**Step 3: Quadtree partitioning.** We define the notion of heavy and light cells. A cell $c \in G_i$ for $i \in [\log(2\Delta)]$ is called *heavy* if $n_c \geq \delta \cdot \frac{\text{OPT}}{2^i}$, and *light* otherwise, where $\delta$ is a parameter fixed in the analysis.

Starting from the coarsest grid $G_{\log(2\Delta)}$, which contains a single cell $[2\Delta]^d$, we recursively refine heavy cells in a quadtree-like manner. Beginning with the entire domain, each heavy cell is subdivided into its $2^d$ children. Whenever a light cell is created, it is added to the collection $\Lambda$ and is not subdivided further. At termination, $\Lambda$ is a partition of $[2\Delta]^d$ consisting exactly of the light cells.

**Step 4: Coreset construction.** Given the partition $\Lambda$, we construct the coreset $S$ as follows. A natural approach is to select a representative for each light cell $c \in \Lambda$ and assign all points in $c$ to this representative. Although this produces

a valid $(k, \epsilon)$-coreset, it fails to satisfy the $\gamma$-resilience requirement. The main issue is that, in $\xi$-close datasets, both the representative and the other points in $c$ may move to different cells, making it impossible to control the resiliency error caused by these movements. Instead, our key idea is to assign points from light cells of $\Lambda$ to sampled representatives drawn from their ancestor cells in the pivot grids.

Specifically, for each light cell $c \in \Lambda$ in a grid $G_i$, we locate the ancestor $c'$ of $c$ in the lowest pivot grid $G_\ell \in \mathcal{G}$ such that $i \leq \ell \leq i + y$, and assign all points in $c$ to the sampled representative chosen for $c'$. See Figure 1 for an illustration.

To this end, we proceed as follows. We initialize $S = \emptyset$ and iterate over each pivot grid $G_\ell \in \mathcal{G}$, where $\ell \in \{j, j - y, j - 2y, \ldots, j - xy\}$. For each cell $c \in G_\ell$, if $c \in \Lambda$, we let $\Lambda_c = c$. Otherwise, we let $\Lambda_c = \Lambda \cap c \cap (G_{\ell-1} \cup \cdots \cup G_{\ell-y+1})$ denote the set of light cells in $\Lambda$ that are contained in $c$ and belong to the non-pivot grids $G_{\ell-1}, \ldots, G_{\ell-y+1}$. Observe that the latter happens if $c$ is a heavy cell and will therefore be partitioned into a set of light cells in Step 3. If $\Lambda_c = \emptyset$, we skip the cell $c$. Otherwise, if $\Lambda_c \cap P \neq \emptyset$, we uniformly sample a point $p_c \in \Lambda_c \cap P$, designate it as the coreset representative $\mathfrak{rep}_c = p_c$, assign it weight $w(\mathfrak{rep}_c) = |\Lambda_c \cap P|$, and add the pair $(\mathfrak{rep}_c, w(\mathfrak{rep}_c))$ to $S$. In this manner, all points in the light cells of $\Lambda_c$ are assigned to $\mathfrak{rep}_c$. Observe that light cells of $\Lambda$ belonging to non-pivot grids $G_{\ell-y-1}, G_{\ell-y-2}, \ldots$ are handled recursively using their corresponding pivot grids. Therefore, they need not be considered when processing the cell $c$ in the pivot grid $G_\ell$.

The main theorem that we prove next is stated as follows.

**Theorem 3.1.** *Let $P \subseteq [0, \Delta]^d$ be a point set, and let $k \in \mathbb{N}$ and $0 < \varepsilon, \xi \leq 1$ be parameters. Then Algorithm RE-SILIENTCORESET returns a $(k, \varepsilon)$-coreset for $k$-median clustering of size $O(\varepsilon^{-d} k \log \Delta + d^{d/2} \varepsilon^{-2d-2} \log^2 \Delta)$ that is $\gamma = O(\frac{\sqrt{d}(1+\xi)}{\log \varepsilon^{-1}})$-resilient for every point set $P' \subseteq [0, \Delta]^d$ that is $\xi$-close to $P$.*

**Guessing** OPT. The coreset construction described above assumes the optimal $k$-median cost OPT is known. This assumption can be eliminated by computing a $(1 \pm \varepsilon)$-approximation to OPT. Because point coordinates are integral, the distance between any two distinct points is at least 1, and the diameter of the point set is $\Delta\sqrt{d} = n^{O(1)}\sqrt{d}$. Thus, the maximum possible $k$-median cost is at most $n\Delta\sqrt{d}$, while the minimum nonzero cost is 1. Hence, we can perform a standard doubling search over $O(\varepsilon^{-1}\log n)$ candidate values to obtain a $(1 \pm \varepsilon)$-approximation of OPT. In practice, we run the algorithm of Theorem 3.1 in parallel for each candidate value of OPT, and select the smallest candidate for which the resulting coreset size is $O(\varepsilon^{-d}k\log\Delta + d^{d/2}\varepsilon^{-2d-2}\log^2\Delta)$.

**$k$-means clustering.** The extension of our $\gamma$-resilient coreset construction to $k$-means clustering, where the goal is to find a set of $k$ centers in $\mathbb{R}^d$ that minimize the sum of squared distances, is straightforward. The only modification is in the definition of a heavy cell. Specifically, a cell $c \in G_i$, for $i \in [\log(2\Delta)]$, is called *heavy* if $n_c \geq \delta \cdot \frac{\text{OPT}}{2^{2i}}$, and *light* otherwise. The remainder of the construction and the analysis proceeds exactly as in the $k$-median case.

### 3.1. Analysis

We split the analysis of our coreset construction into two parts. First, we prove the reported set $S$ is a $(k,\varepsilon)$-coreset of a point set $P \subset [\Delta]^d$ and calculate the size of this coreset. We then show that this coreset is indeed $\gamma$-resilient for $\gamma = \frac{\sqrt{d}(1+\xi)}{\log\varepsilon^{-1}}$ for a constant $a \in \mathbb{N}$. That is, at least $n(1 - \frac{\sqrt{d}(1+\xi)}{\log\varepsilon^{-1}})$ points in two $\xi$-close dataset sets $P, P'$ are assigned to the same coreset points. The missing proofs for this section appear in Appendix A.

**Coreset guarantee.** We first prove that $S$ is a $(k,\varepsilon)$-coreset.

**Lemma 3.2.** *For* $\delta \geq \frac{\varepsilon^{2d+2}}{a\log(2\Delta)4^{2d+1}d^{d/2+1}}$, *the reported set* $S$ *is a* $(k,\varepsilon)$-*coreset for the point set* $P$.

Next, we compute the size of this coreset.

**Lemma 3.3.** *The size of the* $(k,\varepsilon)$-*coreset* $S$ *is* $O(\varepsilon^{-d}k\log\Delta + d^{d/2}\varepsilon^{-2d-2}\log^2\Delta)$.

**Resiliency guarantee.** We now prove that our coreset construction yields a $\gamma$-resilient coreset. Consider two $\xi$-close point sets $P$ and $P'$ together with a bijection $\pi : P \to P'$. We execute the coreset construction independently on $P$ and $P'$, which produces partitions $\Lambda$ and $\Lambda'$, respectively. Importantly, both executions use the same random bits. These random choices fall into three categories.

The first random choice is a translation vector $v \in [0, \Delta]^d$, selected in Step 1, which is applied to both $P$ and $P'$. The

second one is the pivot grid $G_j$, selected uniformly at random from the set of grids $G_{\log(2\Delta)}, \ldots, G_{\log(2\Delta)-y}$, where $y = a\log(1/\epsilon)$. Starting from $G_j$, we construct the set of pivot grids $\mathcal{G} = \{G_j, G_{j-y}, G_{j-2y}, \ldots, G_{j-xy}\}$, which forms an arithmetic progression with initial term $j$ and common difference $-y$. The same pivot grid set $\mathcal{G}$ is used for both $P$ and $P'$. The third source of randomness comes from uniformly sampling coreset representatives from the cells of the pivot grids.

We first consider shifts of the pivot grid set $\mathcal{G}$. Specifically, for each $\ell \in [0..y]$, we define $\mathcal{G}(\ell) = \{G_{j-\ell}, G_{j-y-\ell}, G_{j-2y-\ell}, \ldots, G_{j-xy-\ell}\}$, which is a collection of grids forming an arithmetic progression with initial term $j$ and common difference $-y$, shifted by $\ell$. Let $\mathcal{P}(\ell) = \bigcup_{G_x \in \mathcal{G}(\ell)} \bigcup_{p \in G_x \cap \Lambda}\{p\}$ be the set of points in the light cells of the partition $\Lambda$ that lie in the grids $\mathcal{G}(\ell)$.

The shifted grid sets $\mathcal{G}(\ell)$ partition the grids into $y$ residue classes modulo $y$. Since the initial grid index is chosen uniformly at random, each point falls into each class with probability $1/y$. The lemma below formalizes this balls-in-bins analysis and shows that $\mathcal{P}(\ell)$ contains $n/y$ points in expectation.

**Lemma 3.4.** $\mathbb{E}[|\mathcal{P}(\ell)|] = n/y$.

*Proof.* Let $P$ be the input point set with $|P| = n$. The construction first applies a random shift vector $v$ to the point set, then selects a grid index $j$ uniformly at random from $[y]$. Based on the choice of $j$, the grids are partitioned into $y$ shifted arithmetic progressions $\mathcal{G}(\ell)$ for $\ell \in \{0, 1, \ldots, y-1\}$, according to their indices modulo $y$.

Fix an arbitrary point $p \in P$. After the random shift $v$, the point $p$ lies in some light cell $\mathfrak{c}$ of the partition $\Lambda$. Let $i$ denote the index of the grid $G_i$ that contains this cell $\mathfrak{c}$. Both $\mathfrak{c}$ and $i$ are functions of the random shift $v$.

For a fixed $\ell$, the point $p$ belongs to $\mathcal{P}(\ell)$ if and only if $i \equiv j - \ell \pmod{y}$. We compute $\mathbb{P}[p \in \mathcal{P}(\ell)]$ using the *Law of Total Expectation*. Let $\mathbf{1}_{p \in \mathcal{P}(\ell)}$ be the indicator random variable for the event. Then:

$$\mathbb{P}[p \in \mathcal{P}(\ell)] = \mathbb{E}[\mathbf{1}_{p \in \mathcal{P}(\ell)}] = \mathbb{E}_v\left[\mathbb{E}[\mathbf{1}_{p \in \mathcal{P}(\ell)} \mid v]\right]$$
$$= \mathbb{E}_v\left[\mathbb{P}[p \in \mathcal{P}(\ell) \mid v]\right].$$

Conditioning on $v$ fixes the shift, so $i$ becomes deterministic. Hence, $\mathbb{P}[p \in \mathcal{P}(\ell) \mid v] = \mathbb{P}[i \equiv j - \ell \pmod{y} \mid v]$, where the probability is now over the random choice of $j$.

For any fixed $v$, the value $i$ is a deterministic integer. Because $j$ is chosen uniformly at random from $[y]$, the quantity $j - \ell \mod y$ is uniformly distributed over $\{0, 1, \ldots, y-1\}$. Therefore, $\mathbb{P}[i \equiv j - \ell \pmod{y} \mid v] = \frac{1}{y}$, independently of the specific $i$. Substituting back, $\mathbb{P}[p \in \mathcal{P}(\ell)] = \mathbb{E}_v\left[\frac{1}{y}\right] = \frac{1}{y}$.

Let $X_p$ be the indicator random variable that equals 1 if $p \in \mathcal{P}(\ell)$ and 0 otherwise. From the above, $\mathbb{E}[X_p] = 1/y$ for every point $p \in P$.

Finally, by linearity of expectation, $\mathbb{E}[|\mathcal{P}(\ell)|] = \sum_{p \in P} \mathbb{E}[X_p] = n \cdot \frac{1}{y} = \frac{n}{y}$. Thus, in expectation, the total number of points contained in the light cells of $\Lambda$ that lie in the grids $\mathcal{G}(\ell)$ is exactly $n/y$, as claimed. $\square$

Let $\mathfrak{c} \in \Lambda$ be a light cell in the partition $\Lambda$ and let $P_{\mathfrak{c}}$ denote the set of points contained in this cell. Suppose that $\mathfrak{c} \in G_{j-iy-\ell}$. Let $\pi(P_{\mathfrak{c}})$ be the set of points in $P'$ obtained by applying the bijection $\pi$ to the points in $P_{\mathfrak{c}}$. In the following lemma, we show that $\pi(P_{\mathfrak{c}})$ is cut by the coarser grid $G_{j-iy}$ with probability that decreases exponentially in $\ell$. The initial random shift by the vector $v$ ensures that the position of $\pi(P_{\mathfrak{c}})$ relative to the grid lines is uniformly random, which is crucial for establishing this guarantee.

**Lemma 3.5.** *Suppose that* $\mathfrak{c} \in G_{j-iy-\ell}$. *Then,* $\mathbb{P}\left[\pi(P_{\mathfrak{c}}) \text{ is cut in } G_{j-iy}\right] \leq \frac{\sqrt{d}(1+\xi)}{2^\ell}$.

*Proof.* For each dimension $r \in [d]$, let $diam_r(P_{\mathfrak{c}})$ denote the diameter of $P_{\mathfrak{c}}$ along dimension $r$, and let $diam(P_{\mathfrak{c}})$ denote the Euclidean diameter of $P_{\mathfrak{c}}$. Since all points of $P_{\mathfrak{c}}$ lie in the cell $\mathfrak{c}$, whose side length is $2^{j-iy-\ell}$, we have $diam_r(P_{\mathfrak{c}}) \leq 2^{j-iy-\ell}$ for all $r \in [d]$. Therefore, $diam(P_{\mathfrak{c}}) \leq 2^{j-iy-\ell}\sqrt{d}$.

By Definition 2.1, for any pair of points $p, q \in P$ with $\mathrm{dist}(p,q) > 0$, we have $(1-\xi)\,\mathrm{dist}(p,q) \leq \mathrm{dist}(\pi(p), \pi(q)) \leq (1+\xi)\,\mathrm{dist}(p,q)$. It follows that $diam(\pi(P_{\mathfrak{c}})) \leq (1+\xi)\,diam(P_{\mathfrak{c}}) \leq 2^{j-iy-\ell}\sqrt{d}(1+\xi)$. In particular, for every dimension $r \in [d]$, we have $diam_r(\pi(P_{\mathfrak{c}})) \leq 2^{j-iy-\ell}\sqrt{d}(1+\xi)$.

Let $\mathfrak{c}'$ be the ancestor of $\mathfrak{c}$ in $G_{j-iy}$, whose side length is $2^{j-iy}$. While $P_{\mathfrak{c}} \subseteq \mathfrak{c}'$, the translated set $\pi(P_{\mathfrak{c}})$ may not be fully contained in $\mathfrak{c}'$, since $P'$ may be arbitrarily translated by the adversary before the random shift is applied.

The set $\pi(P_{\mathfrak{c}})$ is cut by the grid $G_{j-iy}$ if it crosses a grid boundary along at least one dimension. For a fixed dimension $r$, the probability of such a crossing is at most $diam_r(\pi(P_{\mathfrak{c}}))/2^{j-iy}$. Thus, $\mathbb{P}\left[\pi(P_{\mathfrak{c}}) \text{ is cut in } G_{j-iy}\right] \leq \frac{2^{j-iy-\ell}\sqrt{d}(1+\xi)}{2^{j-iy}} = \frac{\sqrt{d}(1+\xi)}{2^\ell}$. $\square$

Having Lemmas 3.4 and 3.5 in hand, we now prove that the $(k,\varepsilon)$-coreset returned by Algorithm RESILIENTCORESET for $k$-median clustering is $\gamma = O\left(\frac{\sqrt{d}(1+\xi)}{\log \varepsilon^{-1}}\right)$-resilient. To do so, we bound two sources of error:

**(1) Resiliency of assignments.** A point $p \in P$ can be assigned to a different coreset representative in $P'$ only if the light cell of the partition $\Lambda$ containing $p$ is cut at some coarser grid level. The probability of such a cut decreases exponentially with the shift $\ell$ in the arithmetic progression, because the initial random shift ensures a uniform distribution of points relative to grid boundaries. We combine Lemmas 3.4 and 3.5 to bound this probability, leading to Lemma 3.6, which controls the expected number of points whose assignment changes.

**Lemma 3.6.** *In expectation, the number of points $p \in P$ such that the mapped point $\pi(p)$ in $P'$ is assigned to a coreset representative different from $\pi(\mathfrak{rep}(p))$ is at most $\sqrt{d}(1+\xi)\frac{n}{y}$, where $y = \log(a/\varepsilon)$.*

*Proof.* Let $X$ denote the number of points $p \in P$ for which $\mathfrak{rep}(\pi(p)) \neq \pi(\mathfrak{rep}(p))$, i.e., $X = \sum_{p \in P} \mathbf{1}(\mathfrak{rep}(\pi(p)) \neq \pi(\mathfrak{rep}(p)))$. We will bound $\mathbb{E}[X]$ by analyzing the probability that the points in the light cell of $\Lambda$ containing $p$ are cut at a coarser grid level.

Fix a point $p \in P$, and let $\mathfrak{c} \in \Lambda$ denote the light cell of the partition $\Lambda$ containing $p$. By construction, the representative $\mathfrak{rep}(p)$ is chosen at the coarsest grid level (so-called pivot grid ancestor of the grid that contains $\mathfrak{c}$) in which $\mathfrak{c}$ appears. The mapped point $\pi(p)$ can only be misassigned if the set $\pi(P_{\mathfrak{c}})$ is cut by a coarser grid in the arithmetic progression.

Consider the grids $G_{j-iy-\ell}$ for $\ell \in \{1, \dots, y\}$ and $i \in \{1, \dots, x\}$. Suppose that a cell $\mathfrak{c}$ belongs to the grid $G_{j-iy-\ell}$. Then, by Lemma 3.5, the probability that the point set $\pi(P_{\mathfrak{c}})$ (i.e., the image of $P_{\mathfrak{c}}$ under the mapping to $P'$) is cut by the lines of the pivot grid $G_{j-iy}$ satisfies $\mathbb{P}\left[\pi(P_{\mathfrak{c}}) \text{ is cut in } G_{j-iy}\right] \leq \frac{\sqrt{d}(1+\xi)}{2^\ell}$.

If such a cut occurs, all points in $P_{\mathfrak{c}}$ may be misassigned, contributing to $X$. Using a union bound over all light cells and all relevant grid levels, we obtain

$$\mathbb{E}[X] \leq \sum_{\ell=1}^{y} \sum_{i=1}^{x} \sum_{\mathfrak{c} \in G_{j-iy-\ell} \cap \Lambda} \mathbb{P}\left[\pi(P_{\mathfrak{c}}) \text{ is cut in } G_{j-iy}\right] \cdot |P_{\mathfrak{c}}| ,$$

where recall that $x$ is the largest integer such that $j - xy \geq y$. Applying the bound from Lemma 3.5 gives

$$\mathbb{E}[X] \leq \sqrt{d}(1+\xi) \sum_{\ell=1}^{y} \frac{1}{2^\ell} \sum_{i=1}^{x} \sum_{\mathfrak{c} \in G_{j-iy-\ell} \cap \Lambda} |P_{\mathfrak{c}}| .$$

Observe that for a fixed $\ell$, the inner double sum counts exactly the number of points in $\mathcal{P}(\ell)$, i.e., $\sum_{i=1}^{x} \sum_{\mathfrak{c} \in G_{j-iy-\ell} \cap \Lambda} |P_{\mathfrak{c}}| = |\mathcal{P}(\ell)|$. Taking expectations and using Lemma 3.4, we have $\mathbb{E}[|\mathcal{P}(\ell)|] = \frac{n}{y}$. Substituting this, we obtain $\mathbb{E}[X] \leq \sqrt{d}(1+\xi) \sum_{\ell=1}^{y} \frac{1}{2^\ell} \cdot \frac{n}{y}$. Finally, since $\sum_{\ell=1}^{y} 2^{-\ell} \leq 1$, we conclude $\mathbb{E}[X] \leq \sqrt{d}(1+\xi)\frac{n}{y}$. This establishes that, in expectation, at most $\sqrt{d}(1+\xi)n/y$ points are misassigned, completing the proof. $\square$

**(2) Resiliency of representatives.** During the construction of a coreset for a point set $P$, we select one coreset representative $\mathfrak{rep}_\mathfrak{c}$ uniformly at random for each cell $\mathfrak{c}$ that lies both in a pivot grid and in the partition $\Lambda$. Such a representative $\mathfrak{rep}_\mathfrak{c}$ chosen for $P$ may differ from the representative chosen for the corresponding cell in the input point set $P'$ in two ways. First, the cell $\mathfrak{c}$ may still belong to the partition $\Lambda'$, but the mapped point $\pi(\mathfrak{rep}_\mathfrak{c})$ may no longer lie inside the original cell $\mathfrak{c}$, causing $\mathfrak{c}$ to choose a different representative for $P'$. Second, the cell $\mathfrak{c}$ itself may no longer belong to the partition $\Lambda'$, because the partition induced by the pivot grids may change after mapping points from $P$ to $P'$.

**Case 1: Representative displacement.** The mapped point $\pi(\mathfrak{rep}_\mathfrak{c})$ may no longer lie in the original cell $\mathfrak{c}$. In that case, the cell $\mathfrak{c}$ (if it still belongs to the partition $\Lambda'$ of $P'$) may select a different representative for the points in $P' \cap \mathfrak{c}$ than $\mathfrak{rep}_\mathfrak{c}$. Recall that this can occur because $P$ and $P'$ are $\xi$-close point sets. The following lemma bounds the probability of this event.

**Lemma 3.7.** *Let $S$ be the coreset returned by the construction algorithm for point set $P$, where each representative $\mathfrak{rep}_\mathfrak{c} \in S$ is chosen uniformly at random from $P \cap \mathfrak{c}$ for its corresponding cell $\mathfrak{c}$. Then the expected number of representatives whose image under $\pi$ lies outside their original cell is at most $\sum_{\mathfrak{c} \in \Lambda} \mathbb{P}\left[\pi(\mathfrak{rep}_\mathfrak{c}) \notin \mathfrak{c}\right] \leq \sqrt{d}(1+\xi)\frac{|S|}{y}$.*

The proof of Lemma 3.7 follows the same structure as Lemma 3.6, replacing the sum over all points with a sum over representatives, and using the fact that each representative is a random point from its cell.

**Case 2: Partitioning change.** Second, even if $\pi(\mathfrak{rep}_\mathfrak{c})$ remains in $\mathfrak{c}$, the partition $\Lambda$ changes into $\Lambda'$ after mapping points from $P$ to $P'$ which could be different than the partition $\Lambda$. In particular, a pivot-grid cell may be cut differently, which changes the induced partitioning $\Lambda'$. To bound the expected symmetric difference between the two coresets $S$ and $S'$ that is as a result partitioning change, we need to bound the expected number of partition cells whose assignment changes due to such cuts.

**Lemma 3.8.** *Let $P, P' \subseteq [0..\Delta]^d$ be two point sets such that $P'$ is $\xi$-close to $P$. Let $S$ and $S'$ be the coresets returned by the algorithm of Theorem 3.1 when applied to $P$ and $P'$, respectively. Then, $\mathbb{E}\left[|S \triangle S'|\right] \leq \sqrt{d}(1+\xi)\frac{|S|}{y}$.*

*Proof.* Let $\mathcal{C}$ denote the collection of cells that contribute a representative to the coreset $S$, i.e., the cells that lie both in a pivot grid and in the partition $\Lambda$. Observe that $|\mathcal{C}| = |S|$. For each cell $\mathfrak{c} \in \mathcal{C}$, define the indicator random variable $X_\mathfrak{c}$ to be 1 if the representative associated with $\mathfrak{c}$ changes in $S'$, and 0 otherwise.

A representative associated with $\mathfrak{c}$ can change only if the image $\pi(P_\mathfrak{c})$ is cut by a coarser pivot grid after mapping

points from $P$ to $P'$. Indeed, if no such cut occurs, then the cell structure induced by the pivot grids remains unchanged, and the same representative is preserved.

Now consider a cell $\mathfrak{c}$ that belongs to the grid $G_{j-iy-\ell}$. By Lemma 3.5, the probability that $\pi(P_\mathfrak{c})$ is cut by the pivot grid $G_{j-iy}$ is at most $\frac{\sqrt{d}(1+\xi)}{2^\ell}$. Thus, $\mathbb{E}[X_\mathfrak{c}] \leq \frac{\sqrt{d}(1+\xi)}{2^\ell}$. Summing over all possible offsets $\ell \in \{1, \ldots, y\}$ and using that $\sum_{\ell=1}^{y} \frac{1}{2^\ell} \leq 1$, we obtain $\mathbb{E}[X_\mathfrak{c}] \leq \frac{\sqrt{d}(1+\xi)}{y}$.

Since each changed cell contributes at most one element to the symmetric difference $S \triangle S'$, we have $|S \triangle S'| \leq \sum_{\mathfrak{c} \in \mathcal{C}} X_\mathfrak{c}$. Taking expectations yields $\mathbb{E}\left[|S \triangle S'|\right] \leq \sum_{\mathfrak{c} \in \mathcal{C}} \mathbb{E}[X_\mathfrak{c}] \leq \frac{\sqrt{d}(1+\xi)}{y}|\mathcal{C}|$. Finally, since $|\mathcal{C}| = |S|$, we conclude $\mathbb{E}\left[|S \triangle S'|\right] \leq \sqrt{d}(1+\xi)\frac{|S|}{y}$. $\qquad\square$

# 4. Hardness for Resilient $k$-means Clustering

In this section, we establish the following hardness result for the $k$-means problem under resilience constraints.

**Theorem 4.1.** *For any constants $\alpha > 1$ and $\beta > 1$, and for any (potentially randomized) $\gamma$-resilient $\alpha$-approximation algorithm for $k$-means that opens at most $\beta k$ centers, we have: $\gamma \geq \Omega\left(\frac{\ln(1+\xi)}{\xi\sqrt{\alpha} \cdot \beta \ln(\alpha\beta)}\right)$, even in 1-dimensional space, where $\xi$ denotes the closeness of the input instances.*

We can extend the result to $k$-median and $k$-center and these results are provided in Appendices C and D.

**Theorem 4.2.** *For any constants $\alpha > 1$ and $\beta > 1$, and for any (potentially randomized) $\gamma$-resilient $\alpha$-approximation algorithm for $k$-median that opens at most $\beta k$ centers, we have: $\gamma \geq \Omega\left(\frac{\ln(1+\xi)}{\xi\alpha\beta \ln(\alpha\beta)}\right)$, even in 1-dimensional space, where $\xi$ denotes the closeness of the input instances.*

**Theorem 4.3.** *For any constants $\alpha > 1$ and $\beta > 1$, and for any (potentially randomized) $\gamma$-resilient $\alpha$-approximation algorithm for $k$-center that opens at most $\beta k$ centers, we have: $\gamma \geq \Omega\left(\frac{\ln(1+\xi)}{\xi\alpha\beta \ln(\alpha\beta)}\right)$, even in 1-dimensional space, where $\xi$ denotes the closeness of the input instances.*

## 4.1. Hardness Construction for Resilient $k$-means

We define a family of instances to demonstrate the lower bounds of resilience. While we initially define the instances using a graph metric for clarity, we subsequently show that these can be embedded into 1-dimensional space $\mathbb{R}^1$.

The point set $P$ is organized into $A + 1$ disjoint paths, denoted $P_1, P_2, \ldots, P_{A+1}$. Each path $P_i$ contains $m + 1$ vertices connected by $m$ edges. The total number of points is $n = |P| = (A + 1)(m + 1)$. We assume the distance $\text{dist}(p, q)$ between any two points in different paths $P_i, P_j$ $(i \neq j)$ is arbitrarily large.

We define $A+2$ distinct instances that share the same vertex set and topology but differ in their edge weight functions $w$. For any $p, q \in P_i$, the distance $\mathrm{dist}(p, q)$ is the sum of the edge weights on the unique path between them.

**Instance $J_0$ (Uniform Weights)** In the base instance $J_0$, edge weights are uniform: $w_{J_0}(e) = 1$ for all $e$.

**Instances $J_\ell$ (Targeted Perturbations)** For each $\ell \in \{1, \ldots, A+1\}$, we define a perturbed instance $J_\ell$. In this instance, the weight function $w_{J_\ell}$ scales edges on path $P_\ell$ by a factor $\Lambda$, while weights on all other paths remain 1.

- **On the target path $P_\ell$:** For all $e \in P_\ell$, $w_{J_\ell}(e) = \Lambda$.

- **On all other paths $P_j$ ($j \neq \ell$):** $w_{J_\ell}(e) = 1$.

The formal proof regarding the embedding of this metric into $\mathbb{R}^1$ is provided in Theorem B.1.

**Definition 4.4** (Constraints on Variables). Let $A = c_1 \beta \sqrt{\alpha}$ and $m = k/c_2$ for sufficiently large constant integers $c_1$ and $c_2$, and let $\Lambda = A^{3/2}$.

### 4.2. Analysis

We first establish a lower bound for the $k$-means cost on a single path and bound the cost of the optimum solution for the instances.

**Lemma 4.5.** *Consider a path $P$ of $m$ edges with uniform edge weight $w$. The optimal $k$-means cost of covering this path with $c$ centers, where $1 \leq c < m$, is $\Theta(m^3 w^2 / c^2)$.*

**Lemma 4.6.** *The optimal $k$-means cost for instance $J_0$, denoted $\mathrm{OPT}(J_0)$, satisfies $\mathrm{OPT}(J_0) \in O(m^3 A^3 / k^2)$.*

**Lemma 4.7.** *For any $1 \leq \ell \leq A+1$, the optimal $k$-means cost for instance $J_\ell$ satisfies $\mathrm{OPT}(J_\ell) \in O(m^3 \Lambda^2 / k^2) = O(m^3 A^3 / k^2)$.*

Proofs for these lemmas are provided in Appendix B. We now analyze the resilience of any algorithm on these instances. We show that any $\alpha$-approximate solution must concentrate centers on the target path.

**Lemma 4.8.** *Let $M_\ell$ denote the number of points on path $P_\ell$ assigned to a different center by $\mathcal{A}$ in $J_\ell$ compared to $J_0$. Then, $\sum_{1 \leq \ell \leq A+1} M_\ell \geq \frac{99}{100}|P|$.*

*Proof.* Let $n_\ell$ denote the number of centers that $\mathcal{A}$ opens on the path $P_\ell$ in $J_0$. Observe that $\sum_{1 \leq \ell \leq A+1} n_\ell \leq \beta k$. We compute the number of points that these centers can cover while having a cost below $\alpha OPT(J_\ell)$ in instance $J_\ell$. Notice that each center can cover at most two points by a path of $j$ edges for any value $j$ due to the construct of the instances. Therefore, if the maximum distance of coverage

is set to $y$, in the best case scenario, $\mathcal{A}$ can cover $2 y n_\ell$ points. Similar to the proof of Lemma 4.5, the cost would be $\Omega(n_\ell \Lambda^2 y^3)$. This cost cannot be higher than $\alpha \mathrm{OPT}(J_\ell)$. Therefore, $\Omega(n_\ell \Lambda^2 y^3) \leq \alpha \cdot O(m^3 \Lambda^2 / k^2)$. Adding for $1 \leq \ell \leq A+1$

$$\beta k \Lambda^2 y^3 \leq \alpha(A+1) O(m^3 \Lambda^2 / k^2)$$
$$\beta k y^3 \leq \alpha(A+1) O(m^3 / k^2)$$
$$y^3 \leq \alpha \frac{(A+1)}{\beta} O(m^3 / k^3)$$
$$y^3 \leq \alpha \frac{c_1 \beta \sqrt{\alpha}}{\beta} O(\frac{1}{c_2^3})$$
$$y \leq O(\frac{c_1^{1/3} \sqrt{\alpha}}{c_2}) \leq c_1^{1/3} \sqrt{\alpha},$$

where the last two inequalities follows from Definition 4.4 and the fact that $c_2$ is large enough. Recall that the number of points that these center can cover is $2 n_\ell y$. Therefore, the maximum number of points that the open centers in $J_0$ can cover in the paths $P_\ell$ is at most:

$$2 \beta k c_1^{1/3} \sqrt{\alpha} \leq 2(A+1)(m+1) \frac{c_2}{c_1^{2/3}} \leq |P| / c_1^{1/3},$$

where the first inequality follows from Definition 4.4 and the second follows by setting $c_1$ high enough. Which means that $|P| - |P| / c_1^{1/3}$ are assigned to new centers. The lemma follows by setting $c_1$ high enough. $\square$

**Lemma 4.9.** *For any $1 \leq \ell \leq A+1$, there exists a chain of $\xi$-close instances $J^1, J^2, \ldots, J^t$ such that $J^1 = J_0$, $J^t = J_\ell$, and $t = O\left(\frac{\ln \Lambda}{\ln(1+\xi)}\right)$.*

*Proof.* The maximum edge weight difference factor is $\Lambda$. We construct a sequence where weights change by a factor of $(1+\xi)$ per step. The number of steps $t$ satisfies $(1+\xi)^t \geq \Lambda$, implying $t = O\left(\frac{\ln \Lambda}{\ln(1+\xi)}\right)$. $\square$

**Theorem 4.10.** *Any deterministic $\alpha$-approximation $k$-means algorithm $\mathcal{A}$ that opens at most $\beta k$ centers is $\gamma$-resilient only if: $\gamma \geq \Omega\left(\frac{\ln(1+\xi)}{\xi \sqrt{\alpha} \cdot \beta \ln(\alpha\beta)}\right)$.*

*Proof.* By Lemma 4.8, the total reassignment of points across all $A+1$ instances relative to $J_0$ is $\frac{99}{100}|P|$. Thus, there exists an $\ell$ such that the number of points changing centers between $J_0$ and $J_\ell$ is at least $\frac{99|P|}{100(A+1)} = \Omega(|P|/(A+1))$. By Lemma 4.9, there exist two adjacent $\xi$-close instances in the chain where the number of changed points is at least: $\Omega\left(\frac{|P|/(A+1)}{\frac{\ln \Lambda}{\ln(1+\xi)}}\right) = \Omega\left(\frac{|P|}{\frac{\ln(\alpha\beta)}{\ln(1+\xi)} \sqrt{\alpha}\beta}\right)$, substituting $\Lambda = A^{3/2}$. By Definition 2.2, we conclude $\gamma\xi|P| \geq \Omega\left(\frac{|P|\ln(1+\xi)}{\sqrt{\alpha}\beta \ln(\alpha\beta)}\right)$, which simplifies to the bound. $\square$

## Impact Statement

This paper presents work whose goal is to advance the field of Machine Learning. The nature of our work is mainly theoretical and focus on clustering, and thus its insights might find use in various areas.

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

# A. Missing proofs of Section 3

We first Lemma 3.2 that shows that for $\delta \geq \frac{\varepsilon^{2d+2}}{\log(2\Delta)4^{2d+1}}$, the reported set $S$ is a $(k, \varepsilon)$-coreset for the point set $P$.

*Proof.* Let $C = \{c_1, ..., c_k\} \subset [\Delta]^d$ be an arbitrary set of $k$ centers. For a fixed grid $G_i$, let $\Lambda_i = \Lambda \cap G_i$ be the light cells (of the partitioning $\Lambda$) that are in the grid $G_i$. We partition cells in $\Lambda_i$ into near and far cells with respect to centers in $C$. Let $\Lambda_i^{\text{near}} = \{\mathfrak{c} \in \Lambda_i : \text{dist}(\mathfrak{c}, C) \leq \frac{4\sqrt{d}}{\varepsilon^2}2^i\}$ be the cells of $\Lambda_i$ that are close to the center set $C$ where $\text{dist}(\mathfrak{c}, C) = \min_{c_i \in C} \min_{p \in \mathfrak{c}} \text{dist}(p, c_i)$. Let $\Lambda_i^{\text{far}} = \Lambda_i \setminus \Lambda_i^{\text{near}}$. See Figure 2 for an illustration.

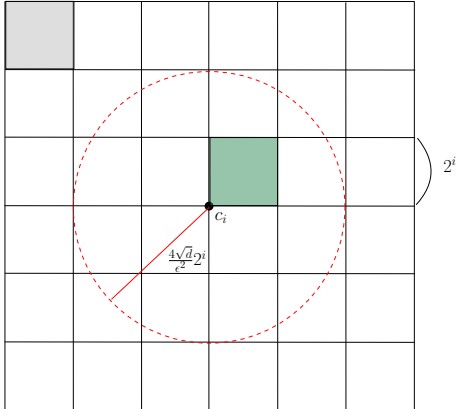

*Figure 2.* The figure specifies a near cell, shown in green, and a far cell, shown in gray, with respect to a center $c_i$.

We next bound the approximation error of far and near cells as follows.

Every point $p$ in a cell $\mathfrak{c} \in \Lambda_i^{\text{far}}$ is charged to its representative within distance at most $a\varepsilon^{-1}2^i\sqrt{d}$ of $p$. To see this, observe that the cell $\mathfrak{c}$ which is in the grid $G_i$ has its ancestor $\mathfrak{c}'$ that is in a pivot grid $G_\ell$ where $i \leq \ell \leq i + y$ where $y = \log(a/\varepsilon)$. Thus, $\ell_{\mathfrak{c}} = 2^i \leq \ell_{\mathfrak{c}'} \leq 2^i 2^y = \frac{a2^i}{\varepsilon}$. All points of the light cell $\mathfrak{c}$ are assigned to a representative point in the ancestor $\mathfrak{c}'$ that is chosen uniformly at random. Thus, the distance between every point in $\mathfrak{c}'$ and its representative point $\mathfrak{rep}_p$ in the ancestor $\mathfrak{c}'$ is at most $\frac{a2^i}{\varepsilon}\sqrt{d}$.

In this way, $\sum_{p \in \mathfrak{c}} \text{dist}(p, \mathfrak{rep}_p) \leq \sum_{p \in \mathfrak{c}} \frac{a2^i}{\varepsilon}\sqrt{d}$. Thus, summarizing over all light cells that are in $\Lambda_i^{\text{far}}$ for all grids $G_\ell$ where $\ell \in [0 \cdots \log 2\Delta]$, we bound the approximation error of these type of cells as follows:

$$\sum_{i=0}^{\log 2\Delta} \sum_{\mathfrak{c} \in \Lambda_i^{\text{far}}} \sum_{p \in \mathfrak{c}} \text{dist}(p, \mathfrak{rep}_p) \leq \sum_{i=0}^{\log 2\Delta} \sum_{\mathfrak{c} \in \Lambda_i^{\text{far}}} \sum_{p \in \mathfrak{c}} \frac{a2^i}{\varepsilon}\sqrt{d} \leq \sum_{i=0}^{\log 2\Delta} \sum_{\mathfrak{c} \in \Lambda_i^{\text{far}}} \sum_{p \in \mathfrak{c}} \frac{a\varepsilon}{4}\text{dist}(p, C) \leq \frac{a\varepsilon}{4}\text{cost}(P, C) \ .$$

Now, we handle the cells that are close (that are in $\Lambda_i^{\text{near}}$). By a volume argument[3], the number of cells of grid $G_i$ (See Figure 2) that are within distance $\frac{4\sqrt{d}}{\varepsilon^2}2^i$ from $C$ is the ratio of the $d$-dimensional ball with the radius $\frac{4\sqrt{d}}{\varepsilon^2}2^i$ to the volume of a hypercube of side length $2^i$ which is $\frac{\frac{\pi^{d/2}}{\Gamma(d/2+1)}\varepsilon^{-2d}(4\sqrt{d})^d 2^{id}}{2^{id}}$, where $\Gamma(d) = (d-1)!$ is the Euler's gamma function that using Stirling's approximation is approximately $\Gamma(d) = \sqrt{2\pi d}(\frac{d}{e})^d$. Thus, the maximum number of such cells is

$$\frac{4^d(d\pi)^{d/2}}{\sqrt{2\pi(d/2+1)}(\frac{d/2+1}{e})^{d/2+1}}\varepsilon^{-2d} \leq (2e\pi d)^{d/2}4^d\varepsilon^{-2d} \leq 4^{2d}\varepsilon^{-2d}d^{d/2} \ .$$

Similar to the points in the far cells, every point $p$ in a cell $\mathfrak{c} \in \Lambda_i^{\text{near}}$ is charged to its representative within distance at most

---

[3]The volume of a $d$-dimensional ball follows a standard formula; see, e.g., Wikipedia's entry on the *Volume of an $n$-ball*[4].

$\text{dist}(p, \mathfrak{rep}_p) \leq \frac{a2^i}{\varepsilon}\sqrt{d}$ of $p$. In addition, every cell $\Lambda_i^{\text{near}}$ is light and therefore, it has less than $\delta\frac{OPT}{2^i}$ points. Thus,

$$\sum_{i=0}^{\log 2\Delta} \sum_{\mathfrak{c}\in\Lambda_i^{\text{near}}} \sum_{p\in\mathfrak{c}} \text{dist}(p, \mathfrak{rep}_p) \leq \sum_{i=0}^{\log 2\Delta} \sum_{\mathfrak{c}\in\Lambda_i^{\text{near}}} \sum_{p\in\mathfrak{c}} \frac{a2^i}{\varepsilon}\sqrt{d} \leq \sum_{i=0}^{\log 2\Delta} \sum_{\mathfrak{c}\in\Lambda_i^{\text{far}}} \frac{a2^i}{\varepsilon}\sqrt{d}\cdot\delta\frac{OPT}{2^i}$$

$$\leq \log(2\Delta)4^{2d}\varepsilon^{-2d}d^{d/2}\frac{a2^i}{\varepsilon}\sqrt{d}\delta OPT \leq \frac{\varepsilon}{4}\text{cost}(P,C) \ ,$$

for $\delta \geq \frac{\varepsilon^{2d+2}}{a\log(2\Delta)4^{2d+1}d^{d/2+1}}$. $\qquad\square$

Next, we prove Lemma 3.3 that shows the size of the $(k,\varepsilon)$-coreset $S$ is $O(\varepsilon^{-d}k\log\Delta + d^{d/2}\varepsilon^{-2d-2}\log^2\Delta)$.

*Proof.* For a fixed grid $G_i$, we let $H_i = \{\mathfrak{c}\in G_i : n_\mathfrak{c} \geq \delta\frac{OPT}{2^i}\}$ and $L_i = G_i\setminus H_i$ be the sets of heavy and light cells of $G_i$, respectively. Let $C^* = \{c_1^*, ..., c_k^*\}\subset[\Delta]^d$ be the optimal set of $k$ centers for the point set $P$. Let $H_i^{\text{near}} = \{\mathfrak{c}\in H_i : \text{dist}(\mathfrak{c}, C^*) \leq \frac{4\sqrt{d}2^i}{\varepsilon}\}$, where $\text{dist}(\mathfrak{c}, C^*) = \min_{c_i\in C^*}\min_{p\in\mathfrak{c}}\text{dist}(p, c_i)$ and $H_i^{\text{far}} = H_i\setminus H_i^{\text{near}}$. Similarly, we define $L_i^{\text{near}} = \{\mathfrak{c}\in L_i : \text{dist}(\mathfrak{c}, C^*) \leq \frac{4\sqrt{d}2^i}{\varepsilon}\}$ and $L_i^{\text{far}} = L_i\setminus L_i^{\text{near}}$.

*Claim A.1.* $|H_i^{\text{near}}\cup L_i^{\text{near}}| \leq 2^{4d}\varepsilon^{-d}k$. Thus, $\sum_{i=0}^{\log(2\Delta)}|H_i^{\text{near}}\cup L_i^{\text{near}}| \leq 2^{4d}\varepsilon^{-d}k\log(2\Delta)$.

*Proof.* Once again, by a volume argument[5], the number of cells of grid $G_i$ that are within distance $4\sqrt{d}2^i\varepsilon^{-1}$ from $C$ is at most $\frac{\frac{\pi^{d/2}}{\Gamma(d/2+1)}\varepsilon^{-d}(4\sqrt{d})^d 2^{id}}{2^{id}} \leq (2e\pi)^{d/2}4^d\varepsilon^{-d} \leq 4^{2d}\varepsilon^{-d}$. Since we have $k$ centers, the maximum number of cells of the grid $G_i$ that are within distance $4\sqrt{d}2^i\varepsilon^{-1}$ of any center in $C^*$ is at most $2^{4d}\varepsilon^{-d}k$. $\qquad\square$

Recall that we say a cell $\mathfrak{c}\in G_i$ is a *heavy* cell if $\mathfrak{c}$ has at least $\delta\frac{OPT}{2^i}$ points. Otherwise, $\mathfrak{c}$ is a *light* cell. A heavy cell $\mathfrak{c}\in G_i$ whose all $2^d$ children (that are in $G_{i-1}$) are light is called a *marked heavy* cell. That is, every child $\mathfrak{c}'$ of $\mathfrak{c}$ has less than $\delta\frac{OPT}{2^{i-1}}$ points.

*Claim A.2.* The number of marked heavy cells that are in $H_i^{\text{far}}$ is at most $\frac{\varepsilon}{4\sqrt{d}\delta}$.

*Proof.* Recall that, for a cell $\mathfrak{c}\in H_i^{\text{far}}$, any point $p\in\mathfrak{c}$ has a distance of at least $\frac{4\sqrt{d}2^i}{\varepsilon}$ to the nearest center of $C^*$. Thus, the total cost of points in $\mathfrak{c}$ to $C^*$ is at least $\frac{4\sqrt{d}2^i}{\varepsilon}\cdot\delta\frac{OPT}{2^i} \geq \frac{4\sqrt{d}\delta}{\varepsilon}OPT$. This essentially means that the number of marked cells that are far from centers of $C^*$ is at most $\frac{\varepsilon}{4\sqrt{d}\delta}$. $\qquad\square$

By combining these bounds, we obtain an upper bound on the size of the coreset. By Claim A.1, we have $\sum_{i=0}^{\log(2\Delta)}|H_i^{\text{near}}\cup L_i^{\text{near}}| \leq 2^{4d}\varepsilon^{-d}k\log(2\Delta)$. Although not all of these cells necessarily belong to $\Lambda_i$, this bound implies that $\sum_{i=0}^{\log(2\Delta)}|\Lambda_i| \leq 2^{4d}\varepsilon^{-d}k\log(2\Delta)$.

Using Claim A.2 and since every marked heavy cell can add $2^d$ light cells to $\Lambda$, the number of light cells in $\Lambda$ whose parents are marked heavy cells is at most

$$\sum_{i=0}^{\log(2\Delta)} 2^d|H_i^{\text{far}}| \leq \log(2\Delta)2^d\frac{\varepsilon}{4\sqrt{d}\delta} \leq \frac{\log^2(2\Delta)4^{2d+1}d^{d/2}}{\varepsilon^{2d+2}} \ ,$$

for $\delta \geq \frac{\varepsilon^{2d+2}}{a\log(2\Delta)4^{2d+1}d^{d/2+1}}$.

The number of coreset points is upper bounded by the number of light cells in $\Lambda$. Thus, in total, we have at most

$$2^{4d}\varepsilon^{-d}k\log(2\Delta) + \log^2(2\Delta)4^{2d+1}d^{d/2}\varepsilon^{-2d-2} = O(\varepsilon^{-d}k\log(\Delta) + \log^2(\Delta)d^{d/2}\varepsilon^{-2d-2})$$

coreset points. $\qquad\square$

---

[5]The volume of a $d$-dimensional ball follows a standard formula; see, e.g., Wikipedia's entry on the *Volume of an n-ball*[6].

# B. Missing Proofs of Section 4

**Theorem B.1.** *The graph structure defined in Section 4 can be embedded into 1-dimensional space $\mathbb{R}^1$.*

*Proof.* Let $v_{i,j}$ denote the $j$-th vertex of the $i$-th path. We define coordinates $f_\ell(v_{i,j}) \in \mathbb{R}^1$ for instance $J_\ell$:

$$f_\ell(v_{i,j}) = i \cdot M + j \cdot w_i^{(\ell)}$$

where $M$ is a sufficiently large separation constant (e.g., $M > \text{OPT}$). The weights $w_i^{(\ell)}$ are:

- In $J_0$: $w_i^{(0)} = 1$ for all $i \in \{1, \ldots, A+1\}$.

- In $J_\ell$: $w_\ell^{(\ell)} = \Lambda$ and $w_i^{(\ell)} = 1$ for all $i \neq \ell$.

$\square$

**Lemma 4.5.** *Consider a path $P$ of $m$ edges with uniform edge weight $w$. The optimal $k$-means cost of covering this path with $c$ centers, where $1 \leq c < m$, is $\Theta(m^3 w^2 / c^2)$.*

*Proof.* For a segment of length $L = mw$, the optimal $k$-means placement partitions $L$ into $c$ sub-segments of length $L/c$, placing a center at each midpoint. The cost for one sub-segment (containing $m/c$ points) is:

$$\sum_{i=1}^{m/2c} 2 \cdot (i \cdot w)^2 = \Theta\left(w^2 \left(\frac{m}{c}\right)^3\right) = \Theta\left(\frac{m^3 w^2}{c^3}\right).$$

Summing over $c$ sub-segments yields a total cost of $\Theta(m^3 w^2 / c^2)$. $\square$

**Lemma 4.6.** *The optimal $k$-means cost for instance $J_0$, denoted $\text{OPT}(J_0)$, satisfies $\text{OPT}(J_0) \in O(m^3 A^3 / k^2)$.*

*Proof.* In $J_0$, we distribute $k$ centers evenly across $A + 1$ paths. Each path receives $c = \Theta(k/(A+1))$ centers. By Lemma 4.5 (with $w = 1$), the cost per path is $\Theta(m^3 (A+1)^2 / k^2)$. Summing over all $A + 1$ paths gives:

$$\text{OPT}(J_0) = (A+1) \cdot O\left(\frac{m^3 A^2}{k^2}\right) = O\left(\frac{m^3 A^3}{k^2}\right).$$

$\square$

**Lemma 4.7.** *For any $1 \leq \ell \leq A + 1$, the optimal $k$-means cost for instance $J_\ell$ satisfies $\text{OPT}(J_\ell) \in O(m^3 \Lambda^2 / k^2) = O(m^3 A^3 / k^2)$.*

*Proof.* To bound the optimal cost $\text{OPT}(J_l)$, we evaluate the cost of a candidate center set C where centers are distributed to prioritize the high-weight target path. We allocate the $k$ centers as follows:

1. Allocate $c_l = k/2$ centers to the target path $P_l$.

2. Distribute the remaining $c_{rem} = k/2$ centers equally among the other $A$ paths. Thus, each path $P_j$ (for $j \neq l$) receives $c_j = \frac{k}{2A}$ centers.

We now compute the resulting costs using the weight function $w_{J_l}$ and Lemma 4.5.

**Cost on the target path $P_l$:** Path $P_l$ has $m$ edges with weight $w_{J_l}(e) = \Lambda$. Using $c_l = k/2$ centers, the cost is:

$$\text{cost}(P_l, \text{C}) = \Theta\left(\frac{m^3 \Lambda^2}{(k/2)^2}\right) = \Theta\left(\frac{m^3 \Lambda^2}{k^2}\right).$$

**Cost on other paths $P_j$ $(j \neq l)$:** Each of the $A$ other paths has uniform edge weights $w_{J_l}(e) = 1$ and is assigned $c_j = \frac{k}{2A}$ centers. The cost for a single such path is:

$$\text{cost}(P_j, \text{C}) = \Theta\left(\frac{m^3(1)^2}{(k/2A)^2}\right) = \Theta\left(\frac{m^3 A^2}{k^2}\right).$$

Summing over all $A$ non-target paths:

$$\sum_{j \neq l} \text{cost}(P_j, \text{C}) = A \cdot \Theta\left(\frac{m^3 A^2}{k^2}\right) = \Theta\left(\frac{m^3 A^3}{k^2}\right).$$

**Total Cost Analysis:** The total $k$-means cost for this allocation is:

$$\text{cost}(P, \text{C}) = \Theta\left(\frac{m^3 \Lambda^2}{k^2}\right) + \Theta\left(\frac{m^3 A^3}{k^2}\right) = \Theta\left(\frac{m^3}{k^2}(\Lambda^2 + A^3)\right).$$

From Definition 4.4, we have $\Lambda = A^{3/2}$, which implies $\Lambda^2 = A^3$. Substituting this in:

$$\text{cost}(P, \text{C}) = \Theta\left(\frac{m^3}{k^2}(A^3 + A^3)\right) = \Theta\left(\frac{m^3 A^3}{k^2}\right).$$

Since $A$ is a constant determined by the parameters $\alpha$ and $\beta$, and $A^3 = \Lambda^2$, we can write this as:

$$\text{OPT}(J_l) \leq \text{cost}(P, \text{C}) \in O\left(\frac{m^3 \Lambda^2}{k^2}\right).$$

This concludes the proof. $\qquad\qquad\qquad\qquad\qquad\qquad\qquad\qquad\qquad\qquad\qquad\qquad\quad$ $\square$

**Theorem 4.1.** *For any constants $\alpha > 1$ and $\beta > 1$, and for any (potentially randomized) $\gamma$-resilient $\alpha$-approximation algorithm for $k$-means that opens at most $\beta k$ centers, we have: $\gamma \geq \Omega\left(\frac{\ln(1+\xi)}{\xi\sqrt{\alpha} \cdot \beta \ln(\alpha\beta)}\right)$, even in 1-dimensional space, where $\xi$ denotes the closeness of the input instances.*

*Proof.* We apply Yao's Minimax Principle (Yao, 1977). It suffices to show that for a distribution $\mathcal{D}$ over input sequences, any deterministic algorithm incurs high expected recourse.

**Input Distribution $\mathcal{D}$.**

1. Select $\ell$ uniformly at random from $\{1, \ldots, A+1\}$.

2. The input is the sequence $\mathcal{S}_\ell = (J^1, \ldots, J^t)$ from Lemma 4.9, transforming $J_0$ into $J_\ell$.

**Deterministic Analysis.** Let $\mathcal{A}$ be a deterministic $\alpha$-approximate algorithm. For $J_0$, it produces a fixed set of centers $C^0$. Let $C_{(0,i)}$ be the number of centers on path $P_i$, where $\sum_{i=1}^{A+1} C_{(0,i)} \leq \beta k$. For the target instance $J_\ell$, let $C_{(\ell,\ell)}$ be the centers on the path $P_\ell$. To be $\alpha$-approximate:

$$\frac{m^3 \Lambda^2}{C_{(\ell,\ell)}^2} \leq \alpha \cdot O\left(\frac{m^3 \Lambda^2}{k^2}\right) \implies C_{(\ell,\ell)} \geq \Omega\left(\frac{k}{\sqrt{\alpha}}\right).$$

Let $N_{req} = \frac{k}{c_2\sqrt{\alpha}}$ be this requirement. The recourse $R(\mathcal{S}_\ell)$ is at least $N_{req} - C_{(0,\ell)}$. The expected recourse over the choice of $\ell$ is:

$$\mathbb{E}[R] \geq \frac{1}{A+1}\sum_{\ell=1}^{A+1}(N_{req} - C_{(0,\ell)}) = N_{req} - \frac{1}{A+1}\sum_{\ell=1}^{A+1}C_{(0,\ell)} \geq N_{req} - \frac{\beta k}{A+1}.$$

With $A = c_1\beta\sqrt{\alpha}$, the second term is strictly smaller than $N_{req}$, yielding $\mathbb{E}[R] \geq \Omega(k/\sqrt{\alpha})$.

**Deriving the Lower Bound on $\gamma$.** If the algorithm is $\gamma$-resilient, the expected recourse per step is at most $\gamma\xi|P|/|P|\cdot k$ (normalized to $k$ for centers). Over $t$ steps:

$$t \cdot \gamma\xi k \geq \mathbb{E}[R] \implies \gamma \geq \frac{\Omega(k/\sqrt{\alpha})}{t\xi k} = \Omega\left(\frac{\ln(1+\xi)}{\xi\sqrt{\alpha}\ln\Lambda}\right).$$

Substituting $\Lambda = A^{3/2} \approx (\beta\sqrt{\alpha})^{3/2}$ yields the final bound:

$$\gamma \geq \Omega\left(\frac{\ln(1+\xi)}{\xi\sqrt{\alpha}\cdot\beta\ln(\alpha\beta)}\right). \qquad \square$$

## C. Hardness for Resilient $k$-median Clustering

In this section, we establish a lower bound for the resilience parameter $\gamma$ for the $k$-median problem. The $k$-median objective function minimizes the sum of Euclidean distances: $\sum_{p\in P}\mathrm{dist}(p, \mathrm{C}^*)$.

**Theorem 4.2.** *For any constants $\alpha > 1$ and $\beta > 1$, and for any (potentially randomized) $\gamma$-resilient $\alpha$-approximation algorithm for $k$-median that opens at most $\beta k$ centers, we have: $\gamma \geq \Omega\left(\frac{\ln(1+\xi)}{\xi\alpha\beta\ln(\alpha\beta)}\right)$, even in 1-dimensional space, where $\xi$ denotes the closeness of the input instances.*

The construction and proof is very similar to the one for $k$-median problem. We define a family of instances using a graph metric that can be embedded into $\mathbb{R}^1$. The construction uses $A + 1$ disjoint paths, $P_1, \ldots, P_{A+1}$, each containing $m + 1$ vertices and $m$ edges. The total number of points is $n = |P| = (A + 1)(m + 1)$. [7]

**Instance $J_0$ (Uniform Weights)** Edge weights are uniform: $w_{J_0}(e) = 1$ for all $e$.

**Instances $J_\ell$ (Targeted Perturbations)** For each $\ell \in \{1, \ldots, A + 1\}$, instance $J_\ell$ is created by scaling the edge weights of the target path $P_\ell$ by a factor $\Lambda$, while leaving other weights as 1.

- **On the target path $P_\ell$:** $w_{J_\ell}(e) = \Lambda$ for $e \in P_\ell$.

- **On all other paths $P_j$ ($j \neq \ell$):** $w_{J_\ell}(e) = 1$.

**Definition C.1** (Constraints for $k$-median)**.** For $k$-median clustering, we set $A = c_1\alpha\beta$ and $m = k/c_2$ for sufficiently large constants $c_1, c_2$. We set the scaling factor $\Lambda = A^2$.

**Lemma C.2.** *The optimal $k$-median cost for a path $P$ of $m$ edges with uniform weight $w$ covered by $c$ centers is $O(m^2w/c)$.*

*Proof of Lemma C.2.* For a segment of length $L = mw$ with $N = m/c$ points, placing a center at the midpoint gives a cost:

$$\sum_{i=1}^{N/2} 2 \cdot (i \cdot w) \leq 4w\frac{(N/2)^2}{2} = O(wN^2) = O\left(\frac{m^2w}{c^2}\right).$$

Summing over $c$ such segments: $c \cdot O(m^2w/c^2) = O(m^2w/c)$. $\qquad \square$

**Lemma C.3.** *The optimal $k$-median costs for instances $J_0$ and $J_\ell$ satisfy:*

$$\mathrm{OPT}(J_0) \in O(m^2A^2/k),$$
$$\mathrm{OPT}(J_\ell) \in O(m^2\Lambda/k) = O(m^2A^2/k).$$

*Proof of Lemma C.3.* For $J_0$, distribute $k$ centers evenly: $c = k/(A+1)$. Total cost $(A+1)\cdot\Theta(m^2/c) = \Theta(m^2A^2/k)$. For $J_\ell$, place $k/2$ centers on $P_\ell$ and $k/2$ on others. Cost on $P_\ell$ is $\Theta(m^2\Lambda/k)$. Cost on others is $A \cdot \Theta(m^2A/k) = \Theta(m^2A^2/k)$. Since $\Lambda = A^2$, the costs are balanced. $\qquad \square$

---

[7] Similar to $k$-means problem, this instance can be embedded into a 1-dimensional space.

**Lemma C.4.** *Let $M_\ell$ be the number of points on path $P_\ell$ assigned to different centers by $\mathcal{A}$ in $J_\ell$ vs $J_0$. Then $\sum_{\ell=1}^{A+1} M_\ell \geq \frac{99}{100}|P|$.*

*Proof.* Let $n_\ell$ be the number of centers $\mathcal{A}$ opens on path $P_\ell$ in the base instance $J_0$. Since the total center budget is $\beta k$, we have $\sum_{\ell=1}^{A+1} n_\ell \leq \beta k$.

In the perturbed instance $J_\ell$, let $x_\ell$ be the number of points on path $P_\ell$ that the algorithm continues to serve using only the $n_\ell$ centers originally placed in $J_0$. By Lemma C.2, the $k$-median cost for serving $x_\ell$ points with $n_\ell$ centers on a path with edge weight $\Lambda$ is $\Omega(\Lambda x_\ell^2/n_\ell)$.

Because $\mathcal{A}$ is an $\alpha$-approximation, this cost cannot exceed $\alpha\mathrm{OPT}(J_\ell)$. Recalling from Lemma C.3 that $\mathrm{OPT}(J_\ell) = O(m^2\Lambda/k)$, we have:

$$\frac{\Lambda x_\ell^2}{n_\ell} \leq \alpha \cdot O\left(\frac{m^2\Lambda}{k}\right) \implies x_\ell^2 \leq O(1) \cdot \frac{\alpha n_\ell m^2}{k} \implies x_\ell \leq O(m)\sqrt{\frac{\alpha n_\ell}{k}}.$$

To find the total number of points across all paths that can be covered by the original $J_0$ centers while remaining $\alpha$-approximate, we sum $x_\ell$ over all $\ell$:

$$\sum_{\ell=1}^{A+1} x_\ell \leq O(m)\sqrt{\frac{\alpha}{k}} \sum_{\ell=1}^{A+1} \sqrt{n_\ell}.$$

Applying the Cauchy-Schwarz inequality ($\sum_{\ell=1}^{A+1}\sqrt{n_\ell} \leq \sqrt{(A+1)\sum_{\ell=1}^{A+1} n_\ell}$), and substituting the total center budget $\sum n_\ell \leq \beta k$:

$$\sum_{\ell=1}^{A+1} x_\ell \leq O(m)\sqrt{\frac{\alpha}{k}}\sqrt{(A+1)\beta k} = O(m)\sqrt{\alpha\beta(A+1)}.$$

The total number of points in the point set $P$ is $|P| = (A+1)m$. We compare the covered points to the total points:

$$\frac{\sum x_\ell}{|P|} \leq \frac{O(m)\sqrt{\alpha\beta(A+1)}}{(A+1)m} = O\left(\sqrt{\frac{\alpha\beta}{A+1}}\right).$$

Substituting the constraint $A = c_1\alpha\beta$, we obtain:

$$\frac{\sum x_\ell}{|P|} \leq O\left(\sqrt{\frac{\alpha\beta}{c_1\alpha\beta}}\right) = O\left(\frac{1}{\sqrt{c_1}}\right).$$

The number of points that must be reassigned is $\sum M_\ell = |P| - \sum x_\ell$. Thus:

$$\sum_{\ell=1}^{A+1} M_\ell \geq |P|\left(1 - \frac{O(1)}{\sqrt{c_1}}\right).$$

By setting $c_1$ to be a sufficiently large constant, the reassignment covers at least $\frac{99}{100}|P|$ points, as required. $\square$

**Theorem C.5.** *Any deterministic $\alpha$-approximate $k$-median algorithm $\mathcal{A}$ with $\beta k$ centers is $\gamma$-resilient only if $\gamma \geq \Omega\left(\frac{\ln(1+\xi)}{\xi\alpha\beta\ln(\alpha\beta)}\right)$.*

*Proof.* By Lemma C.4, there exists $\ell$ such that the number of changed assignments between $J_0$ and $J_\ell$ is $\Omega(|P|/(A+1))$. The chain length between $J_0$ and $J_\ell$ is $t = O(\frac{\ln\Lambda}{\ln(1+\xi)})$. There must be a $\xi$-close step where the number of changes is at least $\frac{|P|}{(A+1)t}$. By the definition of resilience:

$$\gamma\xi|P| \geq \Omega\left(\frac{|P|}{A\cdot t}\right) \implies \gamma \geq \Omega\left(\frac{1}{\xi\alpha\beta\frac{\ln(\alpha\beta)}{\ln(1+\xi)}}\right),$$

substituting $A = \Theta(\alpha\beta)$ and $\Lambda = A^2$. $\square$

The proof of Theorem 4.2 follows by Yao's lemma similar to the proof of Theorem 4.1.

# D. Hardness for Resilient $k$-center Clustering

In this section, we define the $k$-center problem and establish a lower bound for its resilience parameter $\gamma$. Unlike $k$-means and $k$-median, the $k$-center objective is determined solely by the maximum distance, making it inherently more sensitive to perturbations.

**Definition D.1** ($k$-center Objective). Given a point set $P \subset \mathbb{R}^d$ and an integer $k$, the objective of the $k$-center problem is to find a set of $k$ centers $C = \{c_1, \ldots, c_k\}$ that minimizes the maximum distance from any point to its nearest center:

$$\text{cost}(P, C) = \max_{p \in P} \text{dist}(p, C).$$

We denote the optimal $k$-center cost by $\text{OPT}(P, k)$.

**Theorem 4.3.** *For any constants $\alpha > 1$ and $\beta > 1$, and for any (potentially randomized) $\gamma$-resilient $\alpha$-approximation algorithm for $k$-center that opens at most $\beta k$ centers, we have: $\gamma \geq \Omega\left(\frac{\ln(1+\xi)}{\xi \alpha \beta \ln(\alpha\beta)}\right)$, even in 1-dimensional space, where $\xi$ denotes the closeness of the input instances.*

We utilize a family of instances similar to those used in the $k$-means hardness construction. The point set $P$ is organized into $A + 1$ disjoint paths, $P_1, P_2, \ldots, P_{A+1}$. Each path $P_i$ consists of $m$ edges connecting $m + 1$ vertices. The total number of points is $n = |P| = (A + 1)(m + 1)$. We assume the distance $\text{dist}(p, q)$ between any two points in different paths is arbitrarily large (specifically, larger than $2\alpha\text{OPT}$) [8].

**Instance $J_0$ (Uniform Weights)**    In the base instance $J_0$, all edges have uniform unit weight: $w_{J_0}(e) = 1$ for all $e$.

**Instances $J_\ell$ (Targeted Perturbations)**    For each $\ell \in \{1, \ldots, A + 1\}$, we define a perturbed instance $J_\ell$. The weight function $w_{J_\ell}$ scales the target path $P_\ell$ by a factor $\Lambda$, while other paths remain at unit weight.

- **On the target path $P_\ell$:** $w_{J_\ell}(e) = \Lambda$.

- **On all other paths $P_j$ ($j \neq \ell$):** $w_{J_\ell}(e) = 1$.

**Definition D.2** (Constraints on Variables). Let $A = c_1 \beta \alpha$ and $m = k/c_2$ for sufficiently large constant integers $c_1$ and $c_2$, and let $\Lambda = A$. [9]

We first determine the cost (radius) of covering a path in the $k$-center metric.

**Lemma D.3.** *Consider a path $P$ of $m$ edges with uniform edge weight $w$. The optimal $k$-center radius for covering this path with $c$ centers, where $1 \leq c < m$, is $\Theta(mw/c)$.*

*Proof.* An optimal placement partitions the path of length $L = mw$ into $c$ equal segments of length $L/c$, placing a center at the midpoint of each. The maximum radius is $\frac{L}{2c} = \frac{mw}{2c} = \Theta(\frac{mw}{c})$.    □

**Lemma D.4.** *The optimal $k$-center cost for instance $J_0$ satisfies $\text{OPT}(J_0) \in O(mA/k)$.*

*Proof.* In $J_0$, we distribute $k$ centers evenly among the $A + 1$ paths. Each path receives $O(k/A)$ centers. By Lemma D.3 with $w = 1$, the radius is $O(\frac{m}{k/A}) = O(mA/k)$.    □

**Lemma D.5.** *For any $1 \leq \ell \leq A + 1$, the optimal $k$-center cost for instance $J_\ell$ satisfies $\text{OPT}(J_\ell) \in O(m\Lambda/k) = O(mA/k)$.*

*Proof.* We allocate $k/2$ centers to path $P_\ell$ (radius $\Theta(m\Lambda/k)$) and $k/2$ centers to the other $A$ paths (radius $\Theta(mA/k)$). Since $\Lambda = A$, the radii are balanced, and $\text{OPT}(J_\ell) \in O(m\Lambda/k)$.    □

We now show that any $\alpha$-approximate algorithm for $J_\ell$ must concentrate centers on the target path.

---

[8]Similar to $k$-means problem, this instance can be embedded into a 1-dimensional space.
[9]Parameter $\Lambda$ is used for the sake of consistency with previous sections.

**Definition D.6.** Let $C_{(i,j)}$ denote the number of centers the algorithm $\mathcal{A}$ opens on path $P_j$ in instance $J_i$.

**Lemma D.7.** *Any deterministic $\alpha$-approximation $k$-center algorithm $\mathcal{A}$ satisfies $C_{(\ell,\ell)} \in \Omega(k/\alpha)$.*

*Proof.* Let $c$ be the number of centers on $P_\ell$. The radius is at least $\Omega(m\Lambda/c)$. Since $\mathcal{A}$ is $\alpha$-approximate, its radius is at most $\alpha \cdot \text{OPT}(J_\ell) = O(\alpha m\Lambda/k)$. Thus, $\frac{m\Lambda}{c} \leq O\left(\frac{\alpha m\Lambda}{k}\right)$, which implies $c \geq \Omega(k/\alpha)$. $\qquad\square$

**Lemma D.8.** *The sum of the differences in center counts across all instances relative to $J_0$ satisfies:*

$$\sum_{\ell=1}^{A+1}(C_{(\ell,\ell)} - C_{(0,\ell)}) \geq \Omega\left((A+1)\frac{k}{\alpha}\right) - \beta k.$$

*Proof.* By Lemma D.7, for any instance $J_\ell$, an $\alpha$-approximate $k$-center algorithm must open a minimum number of centers on the heavy path $P_\ell$ to satisfy the radius bound. Specifically, there exists a constant $c_{req} > 0$ such that:

$$C_{(\ell,\ell)} \geq c_{req}\frac{k}{\alpha} \quad \text{for all } \ell \in \{1, \dots, A+1\}.$$

Summing this requirement over all $A + 1$ possible perturbed instances, we obtain a lower bound on the total number of centers that would be opened on the target paths if we considered them collectively:

$$\sum_{\ell=1}^{A+1} C_{(\ell,\ell)} \geq \sum_{\ell=1}^{A+1} c_{req}\frac{k}{\alpha} = (A+1)c_{req}\frac{k}{\alpha}.$$

Now consider the base instance $J_0$. The algorithm $\mathcal{A}$ is constrained to open at most $\beta k$ centers in total for any single instance. Let $C_{(0,\ell)}$ be the number of centers $\mathcal{A}$ opens on path $P_\ell$ in instance $J_0$. The sum of these centers across all paths is bounded by the algorithm's budget:

$$\sum_{\ell=1}^{A+1} C_{(0,\ell)} \leq \beta k.$$

We now evaluate the sum of the differences $C_{(\ell,\ell)} - C_{(0,\ell)}$ over all $\ell$:

$$\sum_{\ell=1}^{A+1}(C_{(\ell,\ell)} - C_{(0,\ell)}) = \sum_{\ell=1}^{A+1} C_{(\ell,\ell)} - \sum_{\ell=1}^{A+1} C_{(0,\ell)}$$
$$\geq (A+1)c_{req}\frac{k}{\alpha} - \beta k.$$

Expressing this in asymptotic notation, we have:

$$\sum_{\ell=1}^{A+1}(C_{(\ell,\ell)} - C_{(0,\ell)}) \geq \Omega\left((A+1)\frac{k}{\alpha}\right) - \beta k. \qquad\square$$

**Lemma D.9.** *For any $1 \leq \ell \leq A + 1$, there exists a chain of $\xi$-close instances $J^1, \dots, J^t$ where $J^1 = J_0$, $J^t = J_\ell$, and $t = O\left(\frac{\ln \Lambda}{\ln(1+\xi)}\right)$.*

*Proof.* The instances $J_0$ and $J_\ell$ differ only on the target path $P_\ell$, where edge weights increase from 1 to $\Lambda$. We construct a sequence of intermediate instances by scaling the weights of $P_\ell$ by a factor of at most $(1 + \xi)$ at each step. To reach the total scaling factor of $\Lambda$, the number of steps $t$ must satisfy $(1 + \xi)^t \geq \Lambda$, which implies $t = O\left(\frac{\ln \Lambda}{\ln(1+\xi)}\right)$. $\qquad\square$

**Theorem D.10.** *Any deterministic $\alpha$-approximation $k$-center algorithm $\mathcal{A}$ opening at most $\beta k$ centers is $\gamma$-resilient only if:*

$$\gamma \geq \Omega\left(\frac{\ln(1+\xi)}{\xi\alpha\beta\ln(\alpha\beta)}\right).$$

*Proof.* From Lemma D.8, there exists an $\ell$ such that:

$$C_{(\ell,\ell)} - C_{(0,\ell)} \geq \Omega\left(\frac{k}{\alpha} - \frac{\beta k}{A+1}\right).$$

With $A = c_1 \beta \alpha$, the term $\Omega(\frac{k}{\alpha} - \frac{\beta k}{A+1}) = \Omega(\frac{k}{c_1 \alpha})$ large enough $c_1$. Thus, $C_{(\ell,\ell)} - C_{(0,\ell)} = \Omega(k/\alpha)$. This significant increase in centers on path $P_\ell$ (of size $m$) forces the reassignment of $\Omega(m)$ points to different center identifiers to maintain a valid clustering assignment. Since $|P| = \Theta(A \cdot m)$, this reassignment involves $M_\ell = \Omega(|P|/A)$ points.

By Lemma D.9, this reassignment occurs over a chain of length $t = O(\frac{\ln \Lambda}{\ln(1+\xi)})$. There must exist an instance in the chain where the number of changed assignments is:

$$\frac{M_\ell}{t} = \Omega\left(\frac{|P|}{A \cdot \frac{\ln \Lambda}{\ln(1+\xi)}}\right) = \Omega\left(|P|\frac{\ln(1+\xi)}{\alpha\beta \ln(\alpha\beta)}\right),$$

substituting $A = \Theta(\alpha\beta)$ and $\Lambda = A$. By Definition 2.2, we conclude $\gamma\xi|P| \geq \frac{M_\ell}{t}$, leading to the stated bound. The result for randomized algorithms follows from Yao's Minimax Principle. $\square$

