# Resilient Coresets and Consistent Clustering

## Abstract

Many machine learning problems are geometric at their core, relying on metric representations of data for tasks such as clustering, prototype selection, nearest-neighbor search, and graph-based learning. Furthermore, data is constantly evolving and it is routinely transformed through dimensionality reduction, random projections, feature embeddings, compression, or privacy-preserving mechanisms. These transformations are designed to preserve geometry approximately. As a result, they preserve objective values for many geometric optimization problems, but they fail to guarantee that algorithmic outcomes remain consistent. In this work, we study *resilient data summaries* for geometric optimization. Building on the notion of $\gamma$-*resilient algorithms* from (Ahmadian et al., 2024), we introduce $\gamma$-*resilient coresets*. A $\gamma$-resilient $(k, \varepsilon)$-coreset is a compact, weighted summary that guarantees a $(1 + \varepsilon)$ approximation to the objective and enforces stability at the level of assignments. We complement our positive result with a lower bound showing that to obtain a tight approximation for resilient clustering it is necessary to use a bi-criteria solution.

## 1. Introduction

Many core problems in machine learning are fundamentally geometric. Datasets are often represented as points in a metric space, where distances encode similarity, dissimilarity, or semantic relationships between entities. This geometric perspective underlies a wide range of tasks, including clustering and representation learning, prototype or facility selection, nearest-neighbor search, metric-based classification, and graph or cut problems derived from similarity-based embeddings. In these contexts, the geometry of the data directly shapes both the quality of the objective and the structure of the learned solution, such as cluster assignments, selected representatives, or partitions of the data.

Modern machine learning pipelines rarely operate on a single, fixed representation of the data. Instead, raw data naturally evolves over time (e.g., location data) or is routinely transformed through dimensionality reduction, random projections, feature embeddings, compression, quantization, or privacy-preserving mechanisms. These transformations aim to approximately preserve the underlying geometric structure while improving computational efficiency, memory usage, or robustness. Consequently, the same dataset may appear in multiple representations that are not identical, yet are intended to encode essentially the same metric information. Examples include projecting high-dimensional embeddings into a lower-dimensional space for efficient clustering, sketching or hashing features for scalable nearest-neighbor search, and federated or distributed learning where local updates induce small perturbations to a global dataset.

To formalize this phenomenon, Ahmadian et al. (2024) introduced the notions of $\xi$-*close point sets* and $\gamma$-*resilient algorithms*. Two point sets $P$ and $P'$ are $\xi$-close if there exists a bijection between them that preserves all pairwise distances within a multiplicative factor of $1 \pm \xi$. This notion provides a precise formalization of near-isometric perturbations: the two datasets represent essentially the same geometry, up to small, potentially consequential distortions. A randomized algorithm is $\gamma$-resilient if, when executed independently on two $\xi$-close datasets using *shared randomness*, its outputs differ on at most a $\gamma\xi$ fraction of points in expectation. Unlike classical approximation guarantees, which control only objective values, $\gamma$-resilience enforces pointwise agreement, directly measuring solution stability. Intuitively, this concept captures whether discrete algorithmic decisions vary smoothly with the underlying metric, and whether different representations of the same data lead to nearly identical outputs.

**Our Contributions.** In this work, we extend this framework from individual algorithms to *coresets*. A coreset is a small, weighted summary of a dataset that approximately preserves the objective for a class of optimization problems. Coresets are widely used in scalable machine learning, streaming algorithms, and distributed computation, enabling fast approximations with provable guarantees.

[1]Anonymous Institution, Anonymous City, Anonymous Region, Anonymous Country. Correspondence to: Anonymous Author <anon.email@domain.com>.

Preliminary work. Under review by the International Conference on Machine Learning (ICML). Do not distribute.

We introduce $\gamma$-*resilient* $(k, \varepsilon)$-*coresets*, which simultaneously guarantee $(1 + \varepsilon)$-approximation of the objective and stable assignments. That is, when constructed on two $\xi$-close datasets using shared randomness, the assignment of points to coreset representatives differs on at most a $\gamma\xi$ fraction of points in expectation. In this way, resilience becomes a property of the summary itself, rather than of any particular algorithm.

Achieving $\gamma$-resilience requires careful handling of randomness and discrete decisions. Shared randomness aligns the random choices across executions on different $\xi$-close inputs, ensuring that observed differences arise solely from geometric perturbations rather than independent randomness. However, shared randomness alone is not sufficient: even with perfectly aligned random bits, small metric distortions can still cause comparisons, threshold crossings, or point-to-center assignments to diverge.

To reason rigorously about this, we employ the notion of a *coupling*, which defines a joint probabilistic execution of the algorithm on both inputs. Coupling coordinates the processing of corresponding points so that their computations are as aligned as possible, providing a framework to "track" points under perturbations. Then $\gamma$-resilience can be interpreted as the existence of such a coupling under which all but a $\gamma\xi$ fraction of points follow identical computational paths and are assigned consistently across $\xi$-close datasets. Together, these results establish $\gamma$-resilient coresets as a robust primitive for stable optimization under $\xi$-close perturbations.

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

 $\text{C}^* \subset \mathbb{R}^2$ of $k$ centers that minimizes $\text{cost}(P, \text{C}^*) = \sum_{p \in P} \text{dist}(p, \text{C}^*)$, where $\text{dist}(p, \text{C}^*) = \min_{c \in \text{C}^*} \text{dist}(p, c)$ denotes the Euclidean distance from $p$ to its nearest center in $\text{C}^*$. We denote the optimal $k$-median cost of $P$ by $\text{OPT}(P, k)$, and when $(P, k)$ are clear from context, we simply write OPT.

The optimal clustering induced by the center set $\text{C}^* = \{c_1^*, \ldots, c_k^*\}$ is the partition $\{\mathcal{C}(c_1^*), \ldots, \mathcal{C}(c_k^*)\}$, where each cluster $\mathcal{C}(c_i^*)$ consists of all points that are strictly closer to $c_i^*$ than to any other center in $\text{C}^*$. If a point is equidistant to multiple centers, it may be assigned to any one of them.

Now suppose we have an (approximate or exact) $k$-median algorithm $\mathcal{A}$ whose output is a center set $\text{C} = \{c_1, \ldots, c_k\}$ and a corresponding clustering $\{\mathcal{C}_1, \ldots, \mathcal{C}_k\}$. For any point $p \in P$, we denote by $\text{C}(p)$ the center to which $p$ is assigned

by $\mathcal{A}$. We redefine the output of the algorithm as

$$\mathcal{A}(P, k) = \{(p_1, \mathrm{C}(p_1)), (p_2, \mathrm{C}(p_2)), \ldots, (p_n, \mathrm{C}(p_n))\},$$

where each pair $(p_j, \mathrm{C}(p_j))$ records a point $p_j \in P$ together with its assigned center.

Ahmadian et al. (2024) defines the notions of $\epsilon$-*close* point sets and *resilient clustering*. We recall these definitions in a form adapted to our setting. Each point $p \in P$ is assumed to have a unique identifier, which we denote simply by $p$.

**Definition 2.1** ($\xi$-close point sets (Ahmadian et al., 2024)). Let $0 < \xi \leq 1$, and let $P, P' \subset \mathbb{R}^d$ be point sets with a bijective mapping $\pi : P \to P'$. The sets $P$ and $P'$ are $\xi$-*close* if for all $p, q \in P$ with $\mathrm{dist}(p, q) > 0$, we have

$$(1-\xi)\,\mathrm{dist}(p, q) \ \leq \ \mathrm{dist}(\pi(p), \pi(q)) \ \leq \ (1+\xi)\,\mathrm{dist}(p, q),$$

and if $\mathrm{dist}(p, q) = 0$ then $\mathrm{dist}(\pi(p), \pi(q)) = 0$.

For any two sets $A$ and $B$, we define their symmetric difference as $A \triangle B = (A \setminus B) \cup (B \setminus A)$. We now define $\gamma$-resilient clustering.

**Definition 2.2** ($\gamma$-resilient algorithm (Ahmadian et al., 2024)). An algorithm $\mathcal{A}$ is a $\gamma$-*resilient* $k$-*median algorithm* if for any $\xi$-close point sets $P$ and $P'$ with bijection $\pi : P \to P'$, the outputs satisfy

$$\mathbb{E}[\,|\pi(\mathcal{A}(P, k)) \triangle \mathcal{A}(P', k)|\,] \ \leq \ \gamma\,\xi\,|P|,$$

where the expectation is over the random bits of $\mathcal{A}$ and where we denote the point $(\pi(p), \pi(\mathrm{C}(p)))$ with $\pi(p, \mathrm{C}(p))$, and hence the short-hand $\pi(\mathcal{A}(P, k)) = \{(\pi(p_1), \pi(\mathrm{C}(p_1))), \ldots, (\pi(p_n), \pi(\mathrm{C}(p_n)))\}$.

A key concept introduced in this paper is $\gamma$-*resilient* $(k, \varepsilon)$-*coresets*. We begin by recalling the definition of a $(k, \varepsilon)$-coreset (Har-Peled & Mazumdar, 2004; Frahling & Sohler, 2005) for $k$-median clustering. Given a point set $P \subset \mathbb{R}^d$, a weighted set $S \subset \mathbb{R}^d$ is a $(k, \varepsilon)$-*coreset* for $P$ if, for any set of $k$ centers $\mathrm{C} \subset \mathbb{R}^d$, the $k$-median cost of $S$ approximates that of $P$ within a factor of $1 \pm \varepsilon$, i.e.,

$$(1 - \varepsilon)\,\mathrm{cost}(P, \mathrm{C}) \ \leq \ \mathrm{cost}(S, \mathrm{C}) \ \leq \ (1 + \varepsilon)\,\mathrm{cost}(P, \mathrm{C}),$$

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

 arises from uniformly sampling a representative point from each non-empty cell of every pivot grid.

To establish the resiliency guarantee, we identify two potential sources of error. The first is that two points $p, q \in P$ that lie in the same pivot grid cell may have images

$\pi(p), \pi(q) \in P'$ that do not lie in the corresponding cell. The second source of error is that the images of coreset representatives selected for $P$ may move to different cells in $P'$, causing discrepancies between the assignments of points to representatives in $P$ and in $P'$. In particular, a representative chosen in $P$ may no longer be a valid representative for the corresponding cell in $P'$.

To address the first issue, we prove Lemma 3.4 and Lemma 3.5, which together bound the probability that a pair of points that lie in the same pivot grid cell in $P$ are separated in $P'$. These results are combined in Lemma 3.6 to control the resulting resiliency error. To handle the second issue, Lemma 3.7 bounds the expected number of coreset representatives whose images move to different pivot grid cells in $P'$. We describe these lemmas in detail below; their proofs are deferred to Appendix A.

We first consider shifts of the pivot grid set $\mathcal{G}$. Specifically, for each $\ell \in [0..y]$, we define $\mathcal{G}(\ell) = \{G_{j-\ell}, G_{j-y-\ell}, G_{j-2y-\ell}, \ldots, G_{j-xy-\ell}\}$, which is a collection of grids forming an arithmetic progression with initial term $j$ and common difference $-y$, shifted by $\ell$. Let $\mathcal{P}(\ell) = \bigcup_{G_x \in \mathcal{G}(\ell)} \bigcup_{p \in G_x \cap \Lambda} \{p\}$ be the set of points in the light cells in the partition $\Lambda$ that lie in the grids $\mathcal{G}(\ell)$.

The shifted grid sets $\mathcal{G}(\ell)$ partition the grids into $y$ residue classes modulo $y$. Since the initial grid index is chosen uniformly at random, each point falls into each class with probability $1/y$. The lemma below formalizes this balls in bins analysis and shows that $\mathcal{P}(\ell)$ contains $n/y$ points in expectation.

**Lemma 3.4.** $\mathbb{E}[|\mathcal{P}(\ell)|] = n/y.$

Let $\mathfrak{c} \in \Lambda$ be a light cell in the partition $\Lambda$, and let $P_{\mathfrak{c}}$ denote the set of points contained in this cell. Let $\pi(P_{\mathfrak{c}})$ be the set of points in $P'$ obtained by applying the bijection $\pi$ to the points in $P_{\mathfrak{c}}$. Since $P_{\mathfrak{c}}$ lies in a small cell, its diameter is small, and under the $\xi$-close mapping it can increase by at most a factor of $(1 + \xi)$.

In the following lemma, we show that $\pi(P_{\mathfrak{c}})$ is cut by the coarser grid $G_{j-iy}$ with probability proportional to this diameter relative to the grid cell size, which decreases exponentially in $\ell$. The initial random shift by the vector $v$ ensures that the position of $\pi(P_{\mathfrak{c}})$ relative to the grid lines is uniformly random, which is crucial for establishing this guarantee.

**Lemma 3.5.** *Suppose that $\mathfrak{c} \in G_{j-iy-\ell}$. Then, $\mathbb{P}[\pi(P_{\mathfrak{c}}) \text{ is cut in } G_{j-iy}] \leq \frac{\sqrt{d}(1+\xi)}{2^\ell}.$*

The following lemma proves that our coreset construction is indeed resilient. Intuitively, a point $p \in P$ can only be assigned to a different coreset representative in $P'$ if the set of points in the light cell of the partition $\Lambda$ containing $p$ is cut at some coarser grid level. The probability of such a

cut decreases exponentially with the shift $\ell$ in the arithmetic progression, and the initial random shift ensures a uniform distribution of points relative to grid boundaries.

**Lemma 3.6.** *In expectation, the number of points $p \in P$ such that the mapped point $\pi(p)$ in $P'$ is assigned to a coreset representative different from $\pi(\mathfrak{rep}(p))$ is at most $\sqrt{d}(1 + \xi)\frac{n}{y}$, where $y = \log(a/\varepsilon)$.*

We finally analyze how sensitive the coreset construction is to small perturbations of the input point set. Recall that the construction selects one representative per non-empty partition cell across a collection of pivot grids. As a result, the only way the coresets produced for $P$ and $P'$ can differ is if a partition cell used by the construction is cut differently after mapping points from $P$ to $P'$. Each such cut can affect at most one coreset representative. Therefore, to bound the expected symmetric difference between the two coresets, it suffices to bound the expected number of partition cells whose assignments change due to cutting.

**Lemma 3.7.** *Let $P, P' \subseteq [0..\Delta]^d$ be two point sets such that $P'$ is $\xi$-close to $P$, and let $S$ and $S'$ be the coresets returned by the construction algorithm of Theorem 3.1 when applied to $P$ and $P'$, respectively. Then, $\mathbb{E}\big[|S \triangle S'|\big] \leq \frac{\sqrt{d}(1+\xi)}{y}|S|$, where $y = \log(a/\varepsilon)$ and the expectation is taken over the randomness of the coreset construction.*

# 4. Hardness for Resilient $k$-means Clustering

In this section we prove the following hardness result for the $k$-means problem.

**Theorem 4.1.** *For any constant $1 < \alpha$ and $1 < \beta$, for any (potentiality randomized) $\gamma$-resilient $\alpha$-approximate algorithm for $k$-means that opens at most $\beta k$ centers, we have*

$$\gamma \geq \Omega\left(\frac{\ln(1+\xi)}{\xi\sqrt{\alpha}\,\beta\ln(\alpha\beta)}\right),$$

*even in $1$-dimensional space. Recall that $\xi$ denotes the closeness of the input instances.*

## 4.1. Hardness Construction for Resilient $k$-means

We define a family of instances to demonstrate the lower bounds of consistency. For simplicity, rather than defining the instance in a $1$-dimensional space, we define it as a graph metric. We will show that these instances can be embedded into a $1$-dimensional space $\mathbb{R}^1$.

The point set $P$ is organized into $A + 1$ disjoint paths, denoted $P_1, P_2, \ldots, P_{A+1}$. Each path $P_i$ contains $m$ edges connecting $m + 1$ vertices. The total number of points in the metric space is $n = |P| = (A+1)(m+1)$. We assume the distance $\text{dist}(p, q)$ between any two points in different paths $P_i, P_j$ (where $i \neq j$) is arbitrarily large.

We define $A + 1$ distinct instances that share the same vertex set and topology but differ in their edge weight functions $w$. For any $p, q \in P_i$, the distance $\text{dist}(p, q)$ is the sum of the edge weights on the unique path between them.

**Instance $J_0$ (Uniform Weights)** In the base instance $J_0$, edge weights are assigned one, i.e., $w_{J_0}(e) = 1$ for all $e$.

**Instances $J_l$ (Targeted Perturbations)** For each index $l \in \{1, \ldots, A + 1\}$, we define a perturbed instance $J_l$. In this instance, the weight function $w_{J_l}$ is modified to single out path $P_l$ by applying a large scaling factor, while in the remaining paths, all the edge weights are 1.

- **On the target path $P_l$:** For all edges $e \in P_l$, the weights are set to: $w_{J_l}(e) = \Lambda$.

- **On all other edges on paths $P_j$ $(j \neq l)$:** $w_{J_l}(e) = 1$.

A full formal proof regarding the embedding of this metric is provided in Theorem B.1.

**Definition 4.2. Constraints on the Variables** Let $A = c_1\beta\sqrt{\alpha}$ and $m = k/c_2$ for large constant integers $c_1$ and $c_2$, and let $\Lambda = A^{3/2}$.

## 4.2. Analysis

We first establish a lower bound approximation for the $k$-means cost.

**Lemma 4.3.** *Consider a path $P$ of $m$ edges with uniform edge weight $w$. The optimal $k$-means cost of covering this path with $c$ centers, where $1 \leq c < m$, is $\Theta\left(m^3w^2/c^2\right)$.*

For the proof, please refer to Appendix B.

Let us continue by computing the cost of the optimum solution for the perturbed instances.

**Lemma 4.4.** *The optimal $k$-means cost for instance $J_0$, denoted $\text{OPT}(J_0)$, satisfies $\text{OPT}(J_0) \in O\left(m^3A^3/k^2\right)$.*

**Lemma 4.5.** *For any $1 \leq \ell \leq A + 1$, the optimal $k$-means cost for instance $J_l$ is $\text{OPT}(J_l) \in O\left(m^3\Lambda^2/k^2\right)$.*

The proof of these two lemmas is presented in Appendix B.

We continue to analyze the consistency of any algorithm on these instances. We show that any $\alpha$-approximate solution must concentrate centers on the target path. For the sake of simplicity, we start with the case of deterministic algorithms. Afterwards, we generalize this to non-deterministic algorithms as well.

**Lemma 4.6.** *Let $M_\ell$ denote the number of points on path $P_\ell$ that are assigned to another center by $\mathcal{A}$ in $J_\ell$ compared to $J_0$. We have that*

$$\sum_{1 \leq \ell \leq A+1} M_i \geq \frac{99}{100}|P|$$

*Proof.* Let $n_\ell$ denote the number of centers that $\mathcal{A}$ opens on the path $P_\ell$ in $J_0$. Observe that $\sum_{1 \leq \ell \leq A+1} n_\ell \leq \beta k$. We compute the number of points that these centers can cover while having a cost below $\alpha OPT(J_\ell)$ in instance $J_\ell$. Notice that each center can cover at most two points by a path of $j$ for any value $j$ due to the construct of the instances. Therefore, if the maximum distance of coverage is set to $y$, in the best case scenario, $\mathcal{A}$ can cover $2yn_\ell$ points. Similar to the proof of Lemma 4.3, the cost would be

$$\Omega(n_\ell \Lambda^2 y^3).$$

This cost cannot ne higher than $\alpha OPT(J_\ell)$. Therefore,

$$\Omega(n_\ell \Lambda^2 y^3) \leq O(m^3 \Lambda^2/k^2).$$

Adding for $1 \leq \ell \leq A+1$

$$\beta k \Lambda^2 y^3 \leq (A+1) O(m^3 \Lambda^2/k^2)$$

$$y^3 \leq \frac{(A+1)}{\beta} O(m^3/k^3)$$

$$y \leq O((c_1\sqrt{\alpha})^{1/3} c_2) \leq (c_1\sqrt{\alpha})^{5/12} c_2,$$

where the last two inequalities follows from Definition 4.2 and the fact that $c_1$ is large enough. Recall that the number of points that these center can cover is $2n_\ell y$. Therefore, the maximum number of points that the open centers in $J_0$ can cover in the paths $P_\ell$ is at most:

$$2\beta k (c_1\sqrt{\alpha})^{5/12} c_2 \leq 2(A+1)(m+1)/c_1^{1/12}$$

$$= 2|P|/c_1^{1/12}.$$

Which means that $|P| - |P|/c_1^{1/12}$ are assigned to new centers. The lemma follows by setting $c_1$ high enough. $\square$

**Lemma 4.7.** *For any $1 \leq l \leq A+1$ there exists a chain of $\xi$-close instances $J^1, J^2, \ldots, J^t$ such that $J^1 = J_0$, $J^t = J_l$, and $t = O\left(\frac{\ln \Lambda}{\ln(1+\xi)}\right)$.*

*Proof.* The maximum factor edge weight difference between two instance is $(A+1)\Lambda \leq \Lambda^2$. We can construct a sequence of instances where the weight are changed by a factor of $1+\xi$ at each step. The number of steps $t$ satisfies $(1+\xi)^t \geq \Lambda^2$, implying $t = O\left(\frac{\ln \Lambda}{\ln(1+\xi)}\right)$. $\square$

**Theorem 4.8.** *Consider any deterministic $\alpha$-approximation $k$-means algorithm $\mathcal{A}$ that opens at most $\beta k$ centers. Then the algorithm is $\gamma$-resilient only for*

$$\gamma \geq \Omega\left(\frac{\ln(1+\xi)}{\xi\sqrt{\alpha} \ln(\alpha\beta)}\right).$$

*Proof.* From Lemma 4.6 the total changes of the assignments of points to centers across all $A+1$ instances relative to $J_0$ is $\frac{99}{100}|P|$. Therefore there exists a value $\ell$ such taht the number of the points that have changed their center between the solution in $J_0$ and $J_\ell$ is at least

$$\frac{99|P|}{100(A+1)} = \Omega(P/(A+1)).$$

By Lemma 4.7 we get that there are two $\xi$-close instances that the number of the points that change their center

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

}}\mathrm{dist}(p,c_i)$ and $H_i^{\mathrm{far}} = H_i \setminus H_i^{\mathrm{near}}$. Similarly, we define $L_i^{\mathrm{near}} = \{\mathfrak{c}\in L_i : \mathrm{dist}(\mathfrak{c},C^*) \leq \frac{4\sqrt{d}2^i}{\varepsilon}\}$ and $L_i^{\mathrm{far}} = L_i \setminus L_i^{\mathrm{near}}$.

*Claim* A.1. $|H_i^{\mathrm{near}} \cup L_i^{\mathrm{near}}| \leq 2^{4d}\varepsilon^{-d}k$. Thus, $\sum_{i=0}^{\log(2\Delta)} |H_i^{\mathrm{near}} \cup L_i^{\mathrm{near}}| \leq 2^{4d}\varepsilon^{-d}k\log(2\Delta)$.

*Proof.* Once again, by a volume argument[3], the number of cells of grid $G_i$ that are within distance $4\sqrt{d}2^i\varepsilon^{-1}$ from $C$ is at most $\frac{\frac{\pi^{d/2}}{\Gamma(d/2+1)}\varepsilon^{-d}(4\sqrt{d})^d2^{id}}{2^{id}} \leq (2e\pi)^{d/2}4^d\varepsilon^{-d} \leq 4^{2d}\varepsilon^{-d}$. Since we have $k$ centers, the maximum number of cells of the grid $G_i$ that are within distance $4\sqrt{d}2^i\varepsilon^{-1}$ of any center in $C^*$ is at most $2^{4d}\varepsilon^{-d}k$. □

Recall that we say a cell $\mathfrak{c}\in G_i$ is a *heavy* cell if $\mathfrak{c}$ has at least $\delta\frac{OPT}{2^i}$ points. Otherwise, $\mathfrak{c}$ is a *light* cell. A heavy cell $\mathfrak{c}\in G_i$ whose all $2^d$ children (that are in $G_{i-1}$) are light is called a *marked heavy* cell. That is, every child $\mathfrak{c}'$ of $\mathfrak{c}$ has less than $\delta\frac{OPT}{2^{i-1}}$ points.

*Claim* A.2. The number of marked heavy cells that are in $H_i^{\mathrm{far}}$ is at most $\frac{\varepsilon}{4\sqrt{d}\delta}$.

*Proof.* Recall that, for a cell $\mathfrak{c}\in H_i^{\mathrm{far}}$, any point $p\in\mathfrak{c}$ has a distance of at least $\frac{4\sqrt{d}2^i}{\varepsilon}$ to the nearest center of $C^*$. Thus, the total cost of points in $\mathfrak{c}$ to $C^*$ is at least $\frac{4\sqrt{d}2^i}{\varepsilon}\cdot\delta\frac{OPT}{2^i} \geq \frac{4\sqrt{d}\delta}{\varepsilon}OPT$. This essentially means that the number of marked cells that are far from centers of $C^*$ is at most $\frac{\varepsilon}{4\sqrt{d}\delta}$. □

By combining these bounds, we obtain an upper bound on the size of the coreset. By Claim A.1, we have $\sum_{i=0}^{\log(2\Delta)}|H_i^{\mathrm{near}} \cup L_i^{\mathrm{near}}| \leq 2^{4d}\varepsilon^{-d}k\log(2\Delta)$. Although not all of these cells necessarily belong to $\Lambda_i$, this bound implies that $\sum_{i=0}^{\log(2\Delta)}|\Lambda_i| \leq 2^{4d}\varepsilon^{-d}k\log(2\Delta)$.

Using Claim A.2 and since every marked heavy cell can add $2^d$ light cells to $\Lambda$, the number of light cells in $\Lambda$ whose parents are marked heavy cells is at most

$$\sum_{i=0}^{\log(2\Delta)} 2^d|H_i^{\mathrm{far}}| \leq \log(2\Delta)2^d\frac{\varepsilon}{4\sqrt{d}\delta} \leq \frac{\log^2(2\Delta)4^{2d+1}d^{d/2}}{\varepsilon^{2d+2}}\ ,$$

for $\delta \geq \frac{\varepsilon^{2d+2}}{a\log(2\Delta)4^{2d+1}d^{d/2+1}}$.

The number of coreset points is upper bounded by the number of light cells in $\Lambda$. Thus, in total, we have at most

$$2^{4d}\varepsilon^{-d}k\log(2\Delta) + \log^2(2\Delta)4^{2d+1}d^{d/2}\varepsilon^{-2d-2} = O(\varepsilon^{-d}k\log(\Delta) + \log^2(\Delta)d^{d/2}\varepsilon^{-2d-2})$$

coreset points. □

---

[3]The volume of a $d$-dimensional ball follows a standard formula; see, e.g., Wikipedia's entry on the *Volume of an $n$-ball*[4].

Now, we prove Lemma 3.4 that shows $\mathbb{E}[|\mathcal{P}(\ell)|] = n/y$.

*Proof.* Let $P$ be the input point set with $|P| = n$. Recall that the construction first applies a random shift to the point set and then selects a grid index $j$ uniformly at random from the valid range. Based on this choice, the grids are partitioned into $y$ shifted arithmetic progressions $\mathcal{G}(\ell)$ for $\ell \in \{0, 1, \ldots, y-1\}$, according to their indices modulo $y$.

Fix an arbitrary point $p \in P$, and let $i(p)$ denote the index of the grid $G_{i(p)}$ that contains $p$ after the random shift. For a fixed value of $\ell$, the grid set $\mathcal{G}(\ell)$ consists exactly of those grids whose indices are congruent to $j - \ell$ modulo $y$. Since $j$ is chosen uniformly at random, the residue class of $i(p)$ modulo $y$ is uniformly distributed over $\{0, 1, \ldots, y-1\}$. Consequently, for any fixed $\ell$, the probability that $p$ lies in a grid belonging to $\mathcal{G}(\ell)$ is exactly $1/y$.

Let $X_p$ be the indicator random variable that equals 1 if $p \in \mathcal{P}(\ell)$ and equals 0 otherwise. From the argument above, we obtain $\mathbb{E}[X_p] = 1/y$ for every point $p \in P$.

Finally, applying linearity of expectation over all points in $P$, we have $\mathbb{E}[|\mathcal{P}(\ell)|] = \sum_{p \in P} \mathbb{E}[X_p] = n \cdot \frac{1}{y} = \frac{n}{y}$.

Thus, in expectation, the total number of points contained in the light cells of the partition $\Lambda$ that are in the grids in $\mathcal{G}(\ell)$ is exactly $n/y$, as claimed. $\qquad\square$

Next, we prove Lemma 3.5. This lemma says that for $\mathfrak{c} \in G_{j-iy-\ell}$ $\mathbb{P}\left[\pi(P_{\mathfrak{c}}) \text{ is cut in } G_{j-iy}\right] \leq \frac{\sqrt{d}(1+\xi)}{2^\ell}$.

*Proof.* For each dimension $r \in [d]$, let $diam_r(P_{\mathfrak{c}})$ denote the diameter of $P_{\mathfrak{c}}$ along dimension $r$, and let $diam(P_{\mathfrak{c}})$ denote the Euclidean diameter of $P_{\mathfrak{c}}$. Since all points of $P_{\mathfrak{c}}$ lie in the cell $\mathfrak{c}$, whose side length is $2^{j-iy-\ell}$, we have $diam_r(P_{\mathfrak{c}}) \leq 2^{j-iy-\ell}$ for all $r \in [d]$. Therefore, $diam(P_{\mathfrak{c}}) \leq 2^{j-iy-\ell}\sqrt{d}$.

By Definition 2.1, for any pair of points $p, q \in P$ with $dist(p, q) > 0$, we have $(1 - \xi)\,dist(p, q) \leq dist(\pi(p), \pi(q)) \leq (1+\xi)\,dist(p, q)$. It follows that $diam(\pi(P_{\mathfrak{c}})) \leq (1+\xi)\,diam(P_{\mathfrak{c}}) \leq 2^{j-iy-\ell}\sqrt{d}(1+\xi)$. In particular, for every dimension $r \in [d]$, we have $diam_r(\pi(P_{\mathfrak{c}})) \leq 2^{j-iy-\ell}\sqrt{d}(1 + \xi)$.

Let $\mathfrak{c}'$ be the ancestor of $\mathfrak{c}$ in the grid $G_{j-iy}$, whose side length is $2^{j-iy}$. While $P_{\mathfrak{c}} \subseteq \mathfrak{c}'$, the translated set $\pi(P_{\mathfrak{c}})$ may not be fully contained in $\mathfrak{c}'$, since $P'$ may be arbitrarily translated by the adversary before the random shift is applied.

The set $\pi(P_{\mathfrak{c}})$ is cut by the grid $G_{j-iy}$ only if it crosses a grid boundary along at least one dimension. For a fixed dimension $r$, the probability of such a crossing is at most $diam_r(\pi(P_{\mathfrak{c}}))/2^{j-iy}$. Therefore,

$$\mathbb{P}\left[\pi(P_{\mathfrak{c}}) \text{ is cut in } G_{j-iy}\right] \leq \frac{2^{j-iy-\ell}\sqrt{d}(1+\xi)}{2^{j-iy}} = \frac{\sqrt{d}(1+\xi)}{2^\ell} \quad .$$

$\qquad\square$

Now, we prove Lemma 3.6 that shows in expectation, the number of points $p \in P$ such that the mapped point $\pi(p)$ in $P'$ is assigned to a coreset representative different from $\pi(\mathfrak{rep}(p))$ is at most $\sqrt{d}(1 + \xi)\frac{n}{y}$, where $y = a \log(1/\varepsilon)$.

*Proof.* Let $X$ denote the number of points $p \in P$ for which $\mathfrak{rep}(\pi(p)) \neq \pi(\mathfrak{rep}(p))$, i.e., $X = \sum_{p \in P} \mathbf{1}(\mathfrak{rep}(\pi(p)) \neq \pi(\mathfrak{rep}(p)))$. We will bound $\mathbb{E}[X]$ by analyzing the probability that the points in the light cell of $\Lambda$ containing $p$ are cut at a coarser grid level.

Fix a point $p \in P$, and let $\mathfrak{c} \in \Lambda$ denote the light cell of the partition $\Lambda$ containing $p$. By construction, the representative $\mathfrak{rep}(p)$ is chosen at the coarsest grid level (so-called pivot grid ancestor of the grid that contains $\mathfrak{c}$) in which $\mathfrak{c}$ appears. The mapped point $\pi(p)$ can only be misassigned if the set $\pi(P_{\mathfrak{c}})$ is cut by a coarser grid in the arithmetic progression.

Consider the grids $G_{j-iy-\ell}$ for $\ell \in \{1, \ldots, y\}$ and $i \in \{1, \ldots, x\}$. Suppose that a cell $\mathfrak{c}$ belongs to the grid $G_{j-iy-\ell}$. Then, by Lemma 3.5, the probability that the point set $\pi(P_{\mathfrak{c}})$ (i.e., the image of $P_{\mathfrak{c}}$ under the mapping to $P'$) is cut by the lines of the pivot grid $G_{j-iy}$ satisfies $\mathbb{P}\left[\pi(P_{\mathfrak{c}}) \text{ is cut in } G_{j-iy}\right] \leq \frac{\sqrt{d}(1+\xi)}{2^\ell}$.

If such a cut occurs, all points in $P_{\mathfrak{c}}$ may be misassigned, contributing to $X$. Using a union bound over all light cells and all relevant grid levels, we obtain

$$\mathbb{E}[X] \leq \sum_{\ell=1}^{y} \sum_{i=1}^{x} \sum_{\mathfrak{c} \in G_{j-iy-\ell} \cap \Lambda} \mathbb{P}\left[\pi(P_{\mathfrak{c}}) \text{ is cut in } G_{j-iy}\right] \cdot |P_{\mathfrak{c}}| \; ,$$

where recall that $x$ is the largest integer such that $j - xy \geq y$. Applying the bound from Lemma 3.5 gives

$$\mathbb{E}[X] \leq \sqrt{d}(1+\xi) \sum_{\ell=1}^{y} \frac{1}{2^{\ell}} \sum_{i=1}^{x} \sum_{\mathfrak{c} \in G_{j-iy-\ell} \cap \Lambda} |P_{\mathfrak{c}}| \; .$$

Observe that for a fixed $\ell$, the inner double sum counts exactly the number of points in $\mathcal{P}(\ell)$, i.e.,

$$\sum_{i=1}^{x} \sum_{\mathfrak{c} \in G_{j-iy-\ell} \cap \Lambda} |P_{\mathfrak{c}}| = |\mathcal{P}(\ell)| \; .$$

Taking expectations and using Lemma 3.4, we have $\mathbb{E}[|\mathcal{P}(\ell)|] = \frac{n}{y}$. Substituting this, we obtain

$$\mathbb{E}[X] \leq \sqrt{d}(1+\xi) \sum_{\ell=1}^{y} \frac{1}{2^{\ell}} \cdot \frac{n}{y} \; .$$

Finally, since $\sum_{\ell=1}^{y} 2^{-\ell} \leq 1$, we conclude $\mathbb{E}[X] \leq \sqrt{d}(1+\xi)\frac{n}{y}$. This establishes that, in expectation, at most $\sqrt{d}(1+\xi)n/y$ points are misassigned, completing the proof. $\qquad\square$

We finally prove Lemma 3.7. The statement of this lemma is as follows. Let $P, P' \subseteq [0..\Delta]^d$ be two point sets such that $P'$ is $\xi$-close to $P$, and let $S$ and $S'$ be the coresets returned by the construction algorithm of Theorem 3.1 when applied to $P$ and $P'$, respectively. Then,

$$\mathbb{E}\left[|S \triangle S'|\right] \; \leq \; \frac{\sqrt{d}(1+\xi)}{y} |S| \; ,$$

where $y = a\log(1/\varepsilon)$ and the expectation is taken over the randomness of the coreset construction.

*Proof.* Recall that the coreset construction selects exactly one representative from each non-empty partition cell $\Lambda_c = \Lambda \cap c$ for every cell $c$ of every pivot grid $G_\ell$, where $\ell \in \{j, j-y, j-2y, \ldots, j-xy\}$. Thus, there is a one-to-one correspondence between coreset points and non-empty partition cells used by the construction.

Fix a partition cell $\Lambda_c$ that contributes a representative to $S$. This representative differs between $S$ and $S'$ if and only if the image $\pi(\Lambda_c)$ is cut by the pivot grids, causing the sampled representative for $\Lambda_c$ to change or disappear. Hence, each partition cell can contribute at most one element to the symmetric difference $S \triangle S'$.

By Lemma 3.6, the probability that a given partition cell $\Lambda_c$ is cut is at most $\frac{\sqrt{d}(1+\xi)}{y}$. Let $X_c$ be the indicator random variable that is 1 if $\Lambda_c$ is cut and 0 otherwise. Then, $\mathbb{E}[X_c] \leq \frac{\sqrt{d}(1+\xi)}{y}$.

Since $|S|$ equals the number of non-empty partition cells, the size of the symmetric difference satisfies $|S \triangle S'| \leq \sum_{\Lambda_c} X_c$. By linearity of expectation, we obtain

$$\mathbb{E}\left[|S \triangle S'|\right] \; \leq \; \sum_{\Lambda_c} \mathbb{E}[X_c] \; \leq \; \frac{\sqrt{d}(1+\xi)}{y} |S|,$$

which completes the proof. $\qquad\square$

## B. Missing proofs of Section 4

**Theorem B.1.** *The graph structure defined in Section 4 can be converted to* 1-*deminsional space.*

*Proof.* We demonstrate the construction by embedding the graph vertices into $\mathbb{R}^1$. Let $v_{i,j}$ denote the $j$-th point of the $i$-th path. We define the coordinates $f_l(v_{i,j}) \in \mathbb{R}^1$ for instance $P_l$ as:

$$f_l(v_{i,j}) = i \cdot M + j \cdot w_i^{(l)}$$

where $M$ is a large separation to ensure paths are far away (one can think of it as $\text{OPT} + 1$. The weights $w_i^{(l)}$ are defined as:

- In the base instance $P_0$: $w_i^{(0)} = 1$ for all $i \in \{1, \dots, A+1\}$.

- In the perturbed instance $P_l$: $w_l^{(l)} = \Lambda$ and $w_i^{(l)} = 1$ for all $i \neq l$. □

**Lemma 4.3.** *Consider a path $P$ of $m$ edges with uniform edge weight $w$. The optimal $k$-means cost of covering this path with $c$ centers, where $1 \leq c < m$, is $\Theta\left(m^3 w^2/c^2\right)$.*

*Proof.* If $P$ is a line segment of length $L = mw$ covered by $c$ centers, the optimal placement partitions the line into $c$ segments of length $L/c$ and adding one center in the middle of each segment. The cost for one center would be

$$\Theta(2((w\frac{m}{2c})^2 + (w(\frac{m}{2c} - 1))^2 + \cdots (w(\frac{m}{2c} - \frac{m}{2c}))^2) = \Theta\left(\frac{m^3 w^2}{c^3}\right)$$

Summing these value across the $c$ segments concludes the proof. □

**Lemma 4.4.** *The optimal $k$-means cost for instance $J_0$, denoted $\text{OPT}(J_0)$, satisfies $\text{OPT}(J_0) \in O\left(m^3 A^3/k^2\right)$.*

*Proof.* In instance $J_0$, each of the $A+1$ paths $P_i$ is divided into $k' = k/(A+1)$ pieces. We consider a candidate solution $C$ that places exactly one center at the midpoint of each of the $k'$ pieces in every path $P_i$. Since there are $A+1$ paths, the total number of centers used is $(A+1) \cdot k' = k$.

Under this assignment, no cluster spans across an edge with weight $\Lambda$. Each center $c$ is responsible for a sub-path (piece) of $m'$ vertices with uniform edge weight $w = 1$. By Lemma 4.3, the cost of one such piece is $\Theta((m')^3/1^2) = \Theta((m')^3)$. Since there are $k$ such pieces in total across all paths, the total cost is:

$$\text{OPT}(J_0) \leq \sum_{i=1}^{k} \Theta((m')^3) = \Theta(k \cdot (m')^3)$$

Recalling that $m' = m/k' = m(A+1)/k$, we substitute to find:

$$\text{OPT}(J_0) \in O\left(k \cdot \left(\frac{m(A+1)}{k}\right)^3\right) = O\left(\frac{m^3(A+1)^3}{k^2}\right) = O\left(\frac{m^3 A^3}{k^2}\right). \qquad \square$$

**Lemma 4.5.** *For any $1 \leq \ell \leq A+1$, the optimal $k$-means cost for instance $J_l$ is $\text{OPT}(J_l) \in O\left(m^3 \Lambda^2/k^2\right)$.*

*Proof.* To bound the optimal cost $\text{OPT}(J_l)$, we evaluate the cost of a candidate center set $C$ where centers are distributed to prioritize the high-weight target path. We allocate the $k$ centers as follows:

1. Allocate $c_l = k/2$ centers to the target path $P_l$.

2. Distribute the remaining $c_{rem} = k/2$ centers equally among the other $A$ paths. Thus, each path $P_j$ (for $j \neq l$) receives $c_j = \frac{k}{2A}$ centers.

We now compute the resulting costs using the weight function $w_{J_l}$ and Lemma 4.3.

**Cost on the target path $P_l$:**    Path $P_l$ has $m$ edges with weight $w_{J_l}(e) = \Lambda$. Using $c_l = k/2$ centers, the cost is:

$$\text{cost}(P_l, \text{C}) = \Theta\left(\frac{m^3 \Lambda^2}{(k/2)^2}\right) = \Theta\left(\frac{m^3 \Lambda^2}{k^2}\right).$$

**Cost on other paths $P_j$ ($j \neq l$):**    Each of the $A$ other paths has uniform edge weights $w_{J_l}(e) = 1$ and is assigned $c_j = \frac{k}{2A}$ centers. The cost for a single such path is:

$$\text{cost}(P_j, \text{C}) = \Theta\left(\frac{m^3 (1)^2}{(k/2A)^2}\right) = \Theta\left(\frac{m^3 A^2}{k^2}\right).$$

Summing over all $A$ non-target paths:

$$\sum_{j \neq l} \text{cost}(P_j, \text{C}) = A \cdot \Theta\left(\frac{m^3 A^2}{k^2}\right) = \Theta\left(\frac{m^3 A^3}{k^2}\right).$$

**Total Cost Analysis:**    The total $k$-means cost for this allocation is:

$$\text{cost}(P, \text{C}) = \Theta\left(\frac{m^3 \Lambda^2}{k^2}\right) + \Theta\left(\frac{m^3 A^3}{k^2}\right) = \Theta\left(\frac{m^3}{k^2}(\Lambda^2 + A^3)\right).$$

From Definition 4.2, we have $\Lambda = A^{3/2}$, which implies $\Lambda^2 = A^3$. Substituting this in:

$$\text{cost}(P, \text{C}) = \Theta\left(\frac{m^3}{k^2}(A^3 + A^3)\right) = \Theta\left(\frac{m^3 A^3}{k^2}\right).$$

Since $A$ is a constant determined by the parameters $\alpha$ and $\beta$, and $A^3 = \Lambda^2$, we can write this as:

$$\text{OPT}(J_l) \leq \text{cost}(P, \text{C}) \in O\left(\frac{m^3 \Lambda^2}{k^2}\right).$$

This concludes the proof. $\qquad\square$

**Theorem 4.1.** *For any constant $1 < \alpha$ and $1 < \beta$, for any (potentiality randomized) $\gamma$-resilient $\alpha$-approximate algorithm for $k$-means that opens at most $\beta k$ centers, we have*

$$\gamma \geq \Omega\left(\frac{\ln(1 + \xi)}{\xi\sqrt{\alpha}\,\beta\ln(\alpha\beta)}\right),$$

*even in 1-dimensional space. Recall that $\xi$ denotes the closeness of the input instances.*

*Proof.* We use Yao's Minimax Principle. We start by stating it:

**Lemma B.2** (Yao's Minimax Principle (Yao, 1977))**.** *Let $\Pi$ be a computational problem with a set of input instances $\mathcal{I}$. Let $\mathcal{A}$ be the set of all deterministic algorithms solving $\Pi$. For any randomized algorithm $\mathcal{R}$ and any probability distribution $\mathcal{D}$ over $\mathcal{I}$, the worst-case expected cost is lower-bounded by the expected cost of the optimal deterministic algorithm against $\mathcal{D}$:*

$$\max_{I \in \mathcal{I}} \mathbb{E}[Cost(\mathcal{R}, I)] \geq \min_{A \in \mathcal{A}} \mathbb{E}_{I \sim \mathcal{D}}[Cost(A, I)].$$

By Yao's Minimax Principle, it suffices to construct a distribution $\mathcal{D}$ over input sequences and show that any deterministic algorithm $\mathcal{A}$ incurs a high expected recourse cost against $\mathcal{D}$.

**Input Distribution $\mathcal{D}$.**    We define the distribution as follows:

1. Select an index $l$ uniformly at random from $\{1, \dots, A + 1\}$.

2. The input is the sequence of instances $\mathcal{S}_l = (J^1, \dots, J^t)$ defined in Lemma 4.7, transforming $J_0$ into $J_l$.

Let $\mathcal{A}$ be any deterministic $\alpha$-approximate algorithm that opens at most $\beta k$ centers. We define the Cost function to be the consistency of the algorithm. The theorem follows by applying Yao's Minimax Principle on Theorem 4.8. $\qquad\square$