# OpenReview forum: "Resilient Coresets and Clustering"
_ICML.cc/2026/Conference — ICML 2026 regular_

### Official Review · Reviewer_6oXz · 2026-03-10

**Soundness:** 2
**Presentation:** 3
**Significance:** 3
**Originality:** 3
**Overall Recommendation:** 4
**Confidence:** 4

**Summary:**

The paper addresses the problem of finding resilient data summaries for for geometric optimization, by introducing the new concept of $\gamma$-resilient coresets. The resiliency property ensures that similar coresets are produced on similar inputs. This is motivated by scenarios where multiple representation of the same dataset are used for machine learning tasks, where these representations encode the same information but might not be perfectly identical. Thus, it is desirable that the output of the algorithm should be resilient to such change. The authors and provide an algorithm to compute such resilient coreset, and use this construction to obtain novel approximation algorithms for resilient $k$-median and resilient $k$-means clustering problem, as well as a lower bound for resilient $k$-means.

**Compliance With Llm Reviewing Policy:**

Affirmed.

**Final Justification:**

The rebuttal addressed my concerns, but I maintain my weak accept recommendation. The paper is well written and presents nice theoretical results, but I think that a thourough empirical evaluation against  is needed in order to improve the score to a clear accept.

**Key Questions For Authors:**

- 1: As the authors point out, consistent clustering refers to a related yet distinct area of research, where a consistency property is added on the centers, ensuring the preservation of some centers. However, the title of the paper refers to 'consistent clustering', whereas the main body of the paper defines and discussed 'resilient clustering'. Can the authors clarify the choice of 'consistent clustering' in the title of the paper?
- 2: The authors claim that the presented method is a bi-criteria approximation algorithm, however, there in an approximation guarantee on the cost, number of opened center and resiliency. Should it be a tri-criteria approximation algorithm instead?
- 3: Can the proposed approach be adapted for general $\ell_p$-clustering methods, regarding approximation ratio or hardness?

**Limitations:**

yes

**Strengths And Weaknesses:**

Strengths:
- 1: The paper generalizes the notion of $\gamma$-resiliency presented in Ahmadian et al. to coresets, which is a key component in many applications. $\gamma$-resilient coresets are used in this paper for resilient clustering algorithms, but it opens way for other application on resilient geometric optimization problems.
- 2: The results obtained on resilient clustering directly improve on results presented in Ahmadian et al for $\gamma$-resilient $k$-median and $k$-means clustering.
- 3: The paper is relatively well written and easy to follow.

Weaknesses:
- 1: A natural motivation for the resiliency property, which is missing from the paper, comes from the imprecision inherent in real-world measurements. Consider GPS data as an example: given a ground-truth point set $P_1$​, exact measurement is impossible in practice, and we instead observe a noisy (because of measurement errors) point set $P_2$​. In such settings, a resilient data summary on $P_2$ ensures that we recover a similar gata summary than on (innaccessible) $P_1$.
- 2: Since the proposed approach for $k$-median clustering is extendable to $k$-means clustering, it would be interesting to see if the same approach can work for general $\ell_p$ clustering problem, where the cost function is $\sum_x dist(x,C^*)^p$.
- 3: Similarly, it would be interesting to see if the proposed hardness results carry for $k$-median, or general $\ell_p$ clustering problem.
- 4: An empirical evaluation to complement the theoretical results would be greatly valuable.
- 5: typo l.397 "ne" -> "be".

---

> ### Author Rebuttal · Authors · 2026-03-31
>
> Thank you for your thoughtful review of our paper. Below we provide responses to your questions and comments.
>
>
> ---
>
> > *W1: A natural motivation for the resiliency property … comes from the imprecision inherent in real-world measurements. Consider GPS data …*
>
> Thank you for highlighting this important and natural application. We agree that scenarios involving measurement noise—such as GPS data—provide a compelling motivation for resilience. We will incorporate a similar discussion in the final version of the paper to better illustrate the practical relevance of our definition.
>
>
> ---
>
> > *W2 and Q3: Can the proposed approach be adapted for general $\ell_p$-clustering methods, regarding approximation ratio or hardness?*
>
>
> We believe that our approach can be extended to general $\ell_p$ clustering objectives. However, due to time constraints, we have not yet verified all the technical details required to ensure that the analysis carries through for the general case. We plan to carefully check the proofs, and if the extension holds, we will include it in the camera-ready version.
>
>
> ---
>
> > *W3: Similarly, it would be interesting to see if the proposed hardness results carry for k-median, or general k-clustering problem.*
>
> Great point. The hardness result extends to k-median (the same construction) easily and we will add a corollary about it in the paper. $k$-clustering is more challenging and we leave it to future works.
>
>
> ---
>
> > *W4: An empirical evaluation to complement the theoretical results would be greatly valuable.*
>
> We agree that empirical evaluation would provide valuable insight into the practical behavior of our algorithm. In particular, it would be interesting to benchmark performance on both synthetic and real $\xi$-close datasets, and to study how clustering assignments produced by our coreset compared to those on the original data under natural perturbations such as embeddings, projections, or noise. Such experiments would help quantify how the resilience guarantees translate into stability in practice.
> We view the development of such an experimental framework as an important direction for future work, and plan to explore it in order to better understand the practical impact of our theoretical results.
>
> ---
>
> > *Q1: The title of the paper refers to 'consistent clustering', whereas the main body of the paper defines and discussed 'resilient clustering'. Can the authors clarify the choice of 'consistent clustering' in the title of the paper?*
>
> Great observation, we used consistent and resilient as synonyms referring  to the same concept in this work. We used consistent in the title to avoid using resilient twice.
>
> ---
>
> > *Q2: The authors claim that the presented method is a bi-criteria approximation algorithm, however, there in an approximation guarantee on the cost, number of opened center and resiliency. Should it be a tri-criteria approximation algorithm instead?*
>
> We believe the bi-criteria refers to approximations guarantees and number of the centers regardless of other constraints in the context of k-median, k-means problems. Ahmadian et al. (2024) result that we build on also uses bi-criteria even though they have also an approximation guarantee on the resiliency.
>
>
>
> ---
>
>
> If you feel that our response has adequately addressed your major concerns, we would appreciate it if you could possibly adjust your score accordingly.

---

> > ### Author Rebuttal · Reviewer_6oXz · 2026-04-01
> >
> > Thank you for the response. While I appreciate the clarifications provided, I remain unconvinced by the arguments for answering Q1 and Q2.
> >
> > Q1. Regarding the title, referring to 'consistent' and 'resilient' clustering as the same concept is confusing, as the submission clearly points out that they refer to two distinct problem settings, and the terms should be used with care.
> >
> > Q2. I believe that Ahmadian et al. (2024) should have refered to their result as a tri-criteria approximation, instead of a 'bi-criteria approximation with an additional approximation guarantee over the resiliency'.
> >
> > Moreover, a thourough empirical evaluation to evaluate practical efficiency of the proposed algorithm is needed in order to improve the soundness and significance of the submission.

---

> > > ### Author Response · Authors · 2026-04-07
> > >
> > > Q1. Thank you for the suggestion. We will update the title to "Resilient Coreset and Clustering" in the final version.
> > >
> > > Q2. Thank you for pointing this out. We will precisely define the approximation and clarify the details regarding the tri-criteria in the final version.

---

### Official Review · Reviewer_NNkR · 2026-03-11

**Soundness:** 3
**Presentation:** 3
**Significance:** 2
**Originality:** 2
**Overall Recommendation:** 4
**Confidence:** 4

**Summary:**

The paper introduces γ-resilient $(k,ε)$-coresets, aiming to preserve not only clustering cost but also assignment stability under near-isometric perturbations. The main technical contribution is a randomized grid-based coreset construction for Euclidean $k$-median/$k$-means in constant dimension, together with a hardness result suggesting that resilient clustering inherently requires bi-criteria solutions.

**Compliance With Llm Reviewing Policy:**

Affirmed.

**Final Justification:**

After considering the paper and the rebuttal, I maintain my weak accept recommendation. The paper makes an interesting and technically meaningful contribution through the notion of resilient $(k,\varepsilon)$-coresets, and the rebuttal adequately addressed my main concerns by clarifying the assumptions, the $\xi$-dependence, and the effect of approximating (OPT). Although the scope remains somewhat limited, the rebuttal increased my confidence in the work rather than changing my overall evaluation.

**Key Questions For Authors:**

1. The main theorem is restricted to constant-dimensional Euclidean space on $[\Delta]^d$. How essential are these assumptions? Can any part of the result be extended to higher dimensions, JL-projected spaces, or general metrics?

2. The resilience bound uses $\gamma = O(\sqrt{d(1+\xi} / \log(\epsilon^{-1}))$, which itself depends on $\xi$. Is this dependence inherent to the analysis, or can the guarantee be reformulated in a cleaner way with a $\xi$-independent resilience parameter?

3. The paper states that the $k$-means extension is straightforward. Could the authors clarify which parts of the proof truly carry over unchanged, and whether squared distances introduce any additional technical issues?

4. The lower bound is for resilient clustering algorithms, whereas the upper bound is for resilient coresets. What is the precise algorithmic corollary of the coreset result, and how directly does it match the lower-bound setting?

5. The construction assumes access to OPT and removes this via a doubling search over candidate values. Does this affect the resilience guarantee in any subtle way, especially when two $\xi$-close instances may select slightly different candidate scales?

**Limitations:**

yes

**Strengths And Weaknesses:**

The main strengths are summarized as follows:

1. The paper makes a meaningful conceptual step by extending the notion of resilience from algorithms to coresets. That is an interesting idea, since standard coreset guarantees only preserve objective values, whereas this work also asks for consistency of assignments under perturbations.

2. The resilient coreset construction combines random shifts, hierarchical grids, heavy/light cell decomposition, and pivot-grid representatives in a fairly original way. The main theorem gives an explicit coreset size bound and a resilience guarantee for ξ-close datasets.

3. Beyond the positive result, the paper also proves a hardness result for resilient $k$-means. This strengthens the significance of the problem and helps justify why resilience is fundamentally harder than standard approximation.

The main weakness can be summarized as follows:

1. The main result is for Euclidean k-median/k-means in constant dimension, on a discrete grid $[\Delta]^d$ with $\Delta = n ^{O(1)}$, and it also relies on guessing or approximating $OPT$. This makes the contribution theoretically interesting, but its applicability to datasets with large aspect ratio is limited.

2. The high-level ideas are intuitive, but several delicate points are deferred to the appendix, including the main proof ingredients behind the coreset and resilience guarantees. In particular, the claim that the $k$-means extension is "straightforward" seems to deserve more explanation than just changing the heavy-cell threshold.

3.The high-level ideas are intuitive, but several delicate points are deferred to the appendix, including the main proof ingredients behind the coreset and resilience guarantees. In particular, the claim that the $k$-means extension is "straightforward" seems to deserve more explanation than just changing the heavy-cell threshold.

---

> ### Author Rebuttal · Authors · 2026-03-31
>
> Thank you for your thoughtful review of our paper. Below we provide responses to your questions and comments.
>
> ---
>
> > *W1: The main result is for Euclidean k-media in constant dimension, on a discrete grid $[\Delta]^d$ ....*
>
> We can transform any point set $P \subset \mathbb{R}^d$ into a discrete grid $[\Delta]^d$ with $\Delta = n^{O(1)}$ while preserving the $(1+\epsilon)$-approximation. First, we compute a constant-factor approximation for $k$-median/$k$-means (e.g., KDD’24) to obtain centers $C = {c_1, \dots, c_k}$, and assign points to their nearest centers, forming clusters $P_1, \dots, P_k$. Within each cluster, we translate points so $c_i$ is at the origin and scale by $\alpha = \Theta(n / \epsilon)$. Rounding coordinates to integers then introduces at most an additive $O(\epsilon \cdot \mathrm{OPT})$ error, negligible relative to cluster diameter. The resulting points lie on $[\Delta]^d$, preserving the clustering structure. The scaling ensures small rounding error while keeping coordinates polynomially bounded, giving $\Delta = n^{O(1)}$.
>
> ---
>
> > *W2 and Q3: The high-level ideas are intuitive, but several delicate points are deferred to the appendix … *
>
> Unfortunately we did not have enough space to add more details to the main body of the paper. For the k-means, the rest of the proofs are identical to k-median. We would add the proof for the k-means to the appendix for the full version.
>
>
>
> ---
>
> > *Q1: The main theorem is restricted to constant-dimensional Euclidean space on [\Delta]^d. How essential are these assumptions? ...*
>
> The assumptions of constant dimension and a discrete grid $[\Delta]^d$ are central to our construction. Our approach relies on only a small fraction of points lying near cell boundaries, so misassignments can be controlled. This breaks down in high dimensions, where much of the mass concentrates near boundaries. As a result, many points become sensitive to small perturbations, and for $\xi$-close datasets $P$ and $P'$, the probability of switching cells becomes non-negligible, affecting the partition $\Lambda$ and coreset stability. This makes extending our guarantees to high dimensions challenging.
>
> A natural direction is to use Johnson–Lindenstrauss (JL) projections. While our method could be applied after projection, a standard JL dimension of $O(\log n)$ may still yield large coresets. However, known result (due to Makarychev, Makarychev, Razenshteyn-STOC'19) shows that projection to dimension $O(\log(k/\epsilon)/\epsilon^2)$ preserves clustering costs within $(1\pm\epsilon)$. If $\xi$-closeness is preserved, our approach could yield coresets of size $poly(k,\log\Delta,\epsilon^{-1})$. Extending to this setting or to general metrics remains an open problem.
>
>
> ---
>
> > *Q2: The resilience bound uses $\gamma = O()$ .... Is this dependence inherent to the analysis, ...?*
>
> Let $P$ and $P'$ be two $\xi$-close point sets. The dependence of the resilience parameter $\gamma$ on $\xi$ arises naturally in our analysis, particularly in Lemma 3.5.
>
> For any two points $p,q \in P$ in a cell $c$, we bound the probability that their images $\pi(p), \pi(q) \in P'$ are separated at an ancestor of $c$. This depends on the distance between $\pi(p)$ and $\pi(q)$, which can be as large as $(1+\xi)$ times the diameter of $c$ by definition of $\xi$-closeness. This factor directly propagates into the bound on $\gamma$.
>
> Thus, the dependence on $\xi$ appears inherent to our analysis. We are not aware of a way to remove it or obtain an $\xi$-independent guarantee, which remains an interesting open question.
>
> ---
>
>
> > *Q4: The lower bound is for resilient clustering algorithms, whereas the upper bound is for resilient coresets. ....*
>
> The results are comparable since the coreset can be thought of as a clustering algorithm that opens additional centers. The lower bound also shows that it is hard to improve the results even in a 1-dimensional space even if the algorithm is allowed to open extra centers.
>
> ---
>
> > *Q5: The construction assumes access to OPT and removes this via a doubling search over candidate values. Does this affect the resilience guarantee in any subtle way, ...?*
>
> No, this does not affect the resilience guarantee. Lemmas 3.4–3.7 do not rely on the exact value of $OPT$, so using an approximation via doubling search does not impact the analysis.
>
> Specifically, let $y$ satisfy $OPT \le y \le 2,OPT$. Replacing $OPT$ with $y$ in the thresholds (e.g., $\delta \cdot OPT / 2^i$) changes them by only a constant factor. This may slightly affect the number of selected cells but does not alter the construction or analysis.
>
> Thus, the coreset size increases by at most a constant factor, and since both $\xi$-close instances use comparable approximations, the resilience guarantee remains unchanged up to constant factors.
>
> ---
>
>
> If you feel that our response has adequately addressed your major concerns, we would appreciate it if you could possibly adjust your score accordingly.

---

> > ### Author Rebuttal · Reviewer_NNkR · 2026-04-03
> >
> > Thank you for the detailed response. I will maintain my score at this stage.

---

### Official Review · Reviewer_yd44 · 2026-03-13

**Soundness:** 3
**Presentation:** 2
**Significance:** 3
**Originality:** 2
**Overall Recommendation:** 4
**Confidence:** 4

**Summary:**

This paper addresses the challenge of maintaining solution consistency in geometric optimization under metric perturbations, such as those arising from dimensionality reduction or privacy mechanisms. Overall, the study presents the concept of $\gamma$-resilient $(k, \epsilon)$-coresets, which are compact weighted summaries that guarantee both $(1+\epsilon)$-approximation of the clustering objective and stable point assignments across $\xi$-close datasets. The authors propose a randomized construction using hierarchical grid decompositions and pivot grids to achieve these properties for k-median and k-means clustering. Overall, the authors focus on a central concept of enforcing pointwise assignment consistency as an intrinsic property of the summary itself, complemented by theoretical lower bounds that justify the trade-offs between approximation quality, resilience, and the number of centers used.

**Compliance With Llm Reviewing Policy:**

Affirmed.

**Final Justification:**

Overall, this is a good paper. The rebuttal increased my confidence in the paper’s soundness and practical relevance without changing my original evaluation.

**Key Questions For Authors:**

1. In the process of Step 2 and Step 3, the quadtree partition and gird techniques are applied for dataset to obtain a partition $ \Lambda $. What will happen if the dimensionality d of the dataset is greater than n? Is it still possible to obtain a partition $ \Lambda $?

2. What is the purpose of Step 1(Random shift)? The author should provide a more detailed explanation.

3. Lemma 3.7 bounds the expected number of coreset points that differ between $S$ and $S'$. However, even if a coreset representative $p_c$ is selected in both, their weights $w(rep_c)$ might differ because the underlying light cells in the partition $\Lambda$ might be cut differently. Does your analysis account for the stability of the weights assigned to the coreset points, and how does weight instability affect the $(1+\epsilon)$-approximation?

4. The paper provides strong theoretical bounds on $\gamma$, but there is no empirical evaluation showing how the assignment stability behaves in practice under realistic perturbations (e.g., random projections or privacy noise). Do you have experimental evidence suggesting that the theoretical bound on $\gamma$ is tight, or is it typically much smaller in practice?

**Limitations:**

yes

**Strengths And Weaknesses:**

The strengths can be summarized as follows.
The Lemmas and Theorems are clear. Also, this paper is clearly organized. This paper provides both a randomized construction for k-median and k-means and a lower bound justifying the need for bi-criteria solutions. It introduces pivot grids to overcome the instability of traditional grid-based coresets (where points move across boundaries). And it extends the framework of $\gamma$-resilient algorithms (Ahmadian et al., 2024) to the domain of coresets.

---

> ### Author Rebuttal · Authors · 2026-03-31
>
> Thank you for your thoughtful review of our paper. Below we provide responses to your questions and comments.
>
>
> ---
>
> > *Q1: What will happen if the dimensionality d of the dataset is greater than n? Is it still possible to obtain a partition  $\Lambda$?*
>
> Our construction relies on the assumption that only a small fraction of points lie near cell boundaries, so misassignments can be controlled. This assumption breaks down in high dimensions, where a large fraction of the mass concentrates near boundaries. As a result, many points become sensitive to small perturbations, and for $\xi$-close datasets $P$ and $P'$, the probability of switching cells becomes non-negligible, affecting both the partition $\Lambda$ and coreset stability.
>
> A second challenge is that the number of cells in quadtree partitions grows exponentially with the dimension $d$, increasing both computational cost and instability. While a partition $\Lambda$ can still be constructed even when $d > n$, it becomes less meaningful and less stable, weakening our guarantees.
>
> One possible direction is to avoid explicit spatial partitions and instead work in a metric setting (e.g., KDD’24), which removes explicit dependence on $d$. However, such approaches typically achieve only constant-factor approximations and do not yield $\xi$-resilient $(k,\epsilon)$-coresets. Thus, extending our results to high dimensions remains an important open problem.
>
>
> ---
>
> > *Q2: What is the purpose of Step 1(Random shift)? The author should provide a more detailed explanation.*
>
> The purpose of Step 1 (random shift) is to address boundary instability in the spatial partition. At a high level, preserving $\gamma$-resilience for $\xi$-close point sets $P$ and $P'$ requires ensuring that most points are assigned consistently across the two datasets.
>
> A key challenge is that points lying near cell boundaries in a partition of $P$ may cross into different cells in $P'$ under even small perturbations. This leads to inconsistent assignments and instability in the coreset. The random shift addresses this issue by “randomizing” the placement of cell boundaries, ensuring that only a small fraction of points lie close to any boundary in expectation. This allows us to control the number of points whose assignments may change, and hence bound the induced error.
> We refer to our response to Reviewer etAB (Q1) for a more detailed discussion of the challenges in designing resilient coresets.
>
>
> ---
>
> > *Q3: Lemma 3.7 bounds …Does your analysis account for ..., and how does weight instability affect the $(1+\epsilon)$-approximation?*
>
> In our analysis, the $(1+\epsilon)$-approximation guarantee and resilience are handled separately. Lemma 3.2 shows that $S'$ is a $(k,\epsilon)$-coreset for $P'$, just as $S$ is for $P$, so each independently provides a $(1+\epsilon)$ approximation for its dataset.
>
> Regarding weight stability, even if the same representative $p_c$ appears in both $S$ and $S'$, its weight may differ due to differences in the partitions $\Lambda$ and $\Lambda'$. However, this does not affect the $(1+\epsilon)$ guarantee, since the construction and analysis are applied independently to $P$ and $P'$. Reassignments across cells only change local weight distributions without invalidating the approximation.
>
> Such differences are instead captured by the resilience analysis: changes in cell assignments—and hence weights—contribute to instability between $S$ and $S'$, which is bounded in Lemma 3.7. Thus, weight instability affects resilience, but not the $(1+\epsilon)$ guarantee.
>
> ---
>
> > *Q4: The paper provides strong theoretical bounds on $\lambda$,... Do you have experimental evidence suggesting that the theoretical bound on $\lambda$  is tight ...?*
>
> We thank the reviewer for this insightful question. At present, we do not have empirical evidence evaluating how tight the theoretical bound on $\lambda$ is in practice. Our current work focuses on establishing worst-case guarantees, and the bound on $\lambda$ is derived to hold under all $\xi$-close perturbations, which may be conservative.
>
> We expect that in typical practical settings—such as random projections or moderate noise—the observed instability (and hence the effective $\lambda$) is likely to be significantly smaller than the worst-case bound. Verifying this empirically would be an important next step, as it could provide a more nuanced understanding of how resilience behaves under realistic perturbations.
> A thorough investigation would involve evaluating clustering assignments on both original and perturbed datasets (e.g., under random projections or added noise), and measuring the fraction of points whose assignments change. We consider this an important direction for future work and plan to explore it in a dedicated experimental study.
>
> ---
>
>
> If you feel that our response has adequately addressed your major concerns, we would appreciate it if you could possibly adjust your score accordingly.

---

> > ### Author Rebuttal · Reviewer_yd44 · 2026-04-03
> >
> > Thank you for the detailed response. I prefer to maintain my initial score at this stage.

---

### Official Review · Reviewer_etAB · 2026-03-13

**Soundness:** 4
**Presentation:** 4
**Significance:** 3
**Originality:** 3
**Overall Recommendation:** 4
**Confidence:** 3

**Summary:**

This paper studies clustering under small perturbations of the input metric. In many machine learning pipelines, the data representation may change due to dimensionality reduction, random projections, compression, or other transformations. Although these transformations approximately preserve pairwise distances, they may lead to different clustering assignments even when the clustering objective value remains similar. To address this issue, the authors study the notion of resilient clustering and extend it to the coreset setting.

The paper introduces the concept of a $\gamma$-resilient $(k,\varepsilon)$-coreset. The goal is to construct a small weighted subset of the dataset that simultaneously preserves the clustering objective and maintains stable point-to-center assignments under small metric perturbations. The authors propose a randomized algorithm for constructing such resilient coresets for $k$-median and $k$-means clustering in constant-dimensional Euclidean space. The algorithm is based on a hierarchical grid decomposition with carefully designed pivot grids and shared randomness to control assignment stability. The paper provides theoretical guarantees showing that the constructed coreset achieves both $(1+\varepsilon)$ approximation for the clustering objective and a bounded resilience guarantee. In addition, the authors present a lower bound demonstrating that achieving strong resilience for clustering is inherently difficult, even when allowing a bi-criteria solution.

**Compliance With Llm Reviewing Policy:**

Affirmed.

**Final Justification:**

Thank you for your response. It resolves my concerns.

**Key Questions For Authors:**

1. The paper argues that standard coreset constructions may not satisfy the resilience requirement, but it does not provide a formal impossibility result for existing methods. Can the authors clarify whether any existing coreset algorithms already satisfy the proposed resilience definition?

2. Many modern coreset algorithms are based on sensitivity sampling rather than grid-based constructions. Do the authors expect such approaches to fail under the resilience definition? If so, can they provide intuition or formal arguments explaining why?

3. The proposed algorithm has exponential dependence on the dimension. Do the authors expect the method to be practical for moderate-dimensional datasets commonly encountered in machine learning?

4. Since the resilience definition is motivated by practical data transformations such as embeddings or projections, have the authors considered empirical experiments to evaluate how clustering assignments change under such perturbations in practice?

**Limitations:**

yes

**Strengths And Weaknesses:**

# Strengths

This paper studies an interesting problem by combining the notions of resilient clustering and coresets. The motivation of ensuring clustering stability under small perturbations of the dataset is reasonable and relevant for modern machine learning pipelines where data representations may change.

The paper provides rigorous theoretical analysis and solid proofs for the proposed algorithm. The guarantees for approximation and resilience are clearly defined and formally justified.

The proposed algorithm introduces a non-trivial construction that adapts hierarchical grid-based coreset techniques to enforce resilience. The design of pivot grids and the associated analysis represent a meaningful theoretical contribution.

# Weaknesses

The paper does not formally prove that previous coreset algorithms fail to satisfy the proposed resilience requirement. As a result, it is unclear whether the proposed framework is strictly necessary or whether existing coreset methods could already provide similar guarantees under certain conditions.

The discussion mainly focuses on a specific grid-based coreset construction framework. However, there exist many other coreset generation approaches, such as sensitivity sampling, which are widely used in clustering. The paper does not analyze whether such methods would fail under the resilience definition.

The practical efficiency of the algorithm is unclear. The coreset size has exponential dependence on the dimension, and the algorithm involves multiple hierarchical grids and pivot grid structures, which may limit its applicability in practical high-dimensional settings.

The paper does not include any experimental evaluation. While the theoretical contributions are solid, experiments could help illustrate the practical behavior of the algorithm and provide insights into how the resilience guarantees translate to real datasets.

---

> ### Author Rebuttal · Authors · 2026-03-31
>
> Thank you for your thoughtful review of our paper. Below we provide responses to your questions and comments.
>
> ---
>
> > *W1 and Q1: Would existing coreset methods already provide resiliency guarantees under certain conditions?
> W2 and Q2: Sensitivity sampling based coresets fail under the resilience definition.
> *
>
> Existing coreset constructions do not satisfy $\gamma$-resiliency. Preserving $\gamma$-resiliency for $\xi$-close point sets $P$ and $P'$ requires addressing three fundamental challenges that existing methods fail to handle.
>
> First, *boundary instability*. Points near cell boundaries in $P$ may fall into different cells in $P'$ under small perturbations, causing inconsistent coreset assignments.
>
> Second, *lack of translation invariance*. A fixed spatial partition is not translation-invariant. For example, shifting $P$ to $P'$ can place corresponding points into entirely different cells, breaking consistency.
>
> Third, *oblivious randomness*. Randomness must be fixed in advance and shared between $P$ and $P'$; we cannot adapt it after observing $P$. This severely restricts data-dependent adaptive sampling.
>
> All known coreset constructions violate at least one of these requirements. Cell-based methods (e.g., Frahling \& Sohler, 2005) suffer from boundary instability and translation non-invariance. Our construction addresses these via (1) random shifts to minimize boundary points and (2) random pivot grids to improve translation alignment.
>
> Sensitivity sampling fails primarily due to oblivious randomness. Sensitivities are data-dependent and change significantly under small perturbations, so sampling distributions differ. Even with fixed randomness, the same random bits yield different sampled subsets because the underlying distributions of two $\xi$-close point sets have changed. Thus, sensitivity sampling cannot guarantee $\gamma$-resiliency.
>
> In high dimensions, these challenges become severe. For boundary instability, most points in a high-dimensional cell lie near its boundary, making crossings unavoidable. For translation invariance, the number of cells grows exponentially with dimension, making stability under shifts impossible. Recent metric-based approaches (e.g., KDD'24) avoid explicit partitions but provide only constant-factor approximations, not $\gamma$-resilient $(k,\epsilon)$-coresets.
>
> ---
>
> > *W3, Q3, Q4 and W4: The algorithm’s practical efficiency is unclear due to its exponential dependence on dimension and lack of experiments. Is it practical for moderate dimensions, …*
>
> We believe the algorithm is implementable in practice, particularly for small to moderate-dimensional datasets. Its main components—random shifts, quadtree partitioning, and random pivots—are standard and efficient primitives.
>
> That said, a thorough assessment of practical efficiency requires empirical validation. In particular, benchmarking on synthetic and real $\xi$-close datasets would help evaluate performance in realistic settings. We view developing such an experimental framework as an important direction for future work.
>
> We also note that the coreset size depends exponentially on the dimension due to the cell-based nature of our construction. As discussed in our response to Q1, designing $\xi$-resilient $(k,\epsilon)$-coresets without this dependence appears challenging. Thus, while our method is suitable for low to moderate dimensions, extending it to high dimensions remains an important open problem.
>
> We agree that experiments would provide insight into how resilience translates to real data. In particular, evaluating clustering stability under perturbations such as embeddings, projections, or noise would help quantify practical robustness.
>
> Furthermore, constant dimension setting is a very common setting in k-means / k-median clustering problems. Here are a few examples of accepted papers in ICML and NeurIPS:
>
>
> [1] "Sensitivity Sampling for Coreset-Based Data Selection", "ICML, 2024".
>
> [2] "Perturb-and-Project: Differentially Private Similarities and Marginals", "ICML, 2024".
>
> [3] "Coresets for Clustering with Missing Values", "NeurIPS, 2021".
>
> [4] "Explainable k-Means and k-Medians Clustering", "ICML, 2020".
>
> Finally, we emphasize that the primary focus of this paper is theoretical. It is common in the literature (including several ICML works below ) for papers on clustering and coresets to emphasize theoretical contributions without accompanying experiments. Nonetheless, we view empirical validation as an important next step and plan to pursue it in future work.
>
> [5] "Explainable k-Means and k-Medians Clustering", "ICML, 2020".
>
> [6] "Near-optimal Algorithms for Explainable k-Medians and k-Means", "ICML, 2021".
>
> [7] "The Complexity of k-Means Clustering when Little is Known", "ICML, 2022".
>
> ---
>
>
> If you feel that our response has adequately addressed your major concerns, we would appreciate it if you could possibly adjust your score accordingly.

---

> > ### Author Rebuttal · Reviewer_etAB · 2026-04-03
> >
> > Thank you for your response. It resolves my concerns.

---

### Decision · Program_Chairs · 2026-04-30

**Decision:**

Accept (regular)

**Comment:**

This is a pure theory paper. It successfully combines two concepts related to clustering: resilience (stable point-to-center allocation under small input perturbations) and coresets (small subsets of dataset whose optimal clustering induces a near-optimal clustering of the entire dataset).

The paper is clear and the theory is sound. The reviewers appreciate that the construction is nontrivial and they find the proofs interesting.

Some weakness of the paper is that it contains no experiments even though the problem is relevant to practice, and one can actually be worried that the proposed algorithm would not work very well in practice because of the exponential dependence on dimension.